

# A new fractal-theory-based criterion for hydrological model calibration

Zhixu Bai[1], Yao Wu[1], Di Ma[1], Yue-Ping Xu[1]

[1]Institute of Hydrology and Water Resources, Civil Engineering, Zhejiang University, Hangzhou,
310058, China

*Correspondence to:* Yue-Ping Xu (*yuepingxu@zju.edu.cn*)

**Abstract.** Fractality has been found in many areas and has been used to describe the internal features of time series. But is it possible to use fractal theory to improve the performance of hydrological models? This study aims to investigate the potential benefits of applying fractal theory in model calibration. A new criterion named ratio of fractal dimensions (RD) is defined as the ratio of fractal dimensions of simulated and observed streamflow series. To combine the advantages of fractal theory with classical criteria based on squared residuals, a multi-objective calibration strategy is designed. The selected classical criterion is Nash-Sutcliffe efficiency (E). The E-RD strategy is tested in three study cases with different climate and geography. The results of experiment reveal that, from most aspects, introducing RD into model calibration makes the simulation of streamflow components more reasonable. Besides, in calibration, only little decrease of E occurs when pursuing better RD. We therefore recommend choosing the best E among the parameter sets whose RD is around 1.

Key words: fractal theory; Hausdorff dimension; multi-objective calibration; hydrological model

## 1 Introduction

Since firstly introduced by Hurst in 1951, fractality of streamflow series has been studied for decades (Hurst, 1951). There has been a spectacular growth in fractal theory, which was expended to various areas and to multifractal theory (Bai et al., 2019; Davis et al., 1994). The fractality of time series is generally considered as a reflection of self-affinity, periodicity, long-term memory and irregularity (Bai et al., 2019; Hurst, 1951; Mandelbrot, 2004). However, applications of fractal theory in hydrology were limited in simple streamflow analysis, mostly only using the Hurst index (Katsev and Lheureux, 2003). Also, some literatures mentioned other indexes based on fractal theory (Bai et al., 2019; Yu et



al., 2014; Zhang et al., 2010), but again, the research objects were only observed hydrological data.

Recent studies made a progress to hydrological modelling based on fractal theory (Zhang et al., 2010),

but the model only reconstructs flood/drought grades series. As demonstrated by all these literatures,

the fractality of observed streamflow series (as well as other hydro-meteorological data) is inherent

and represents some peculiarity of their study cases. However, few studies have tried to explore the

applications of fractal theory in hydrological model calibration. To our best knowledge, the only

exception is Onyutha et al. (2019), who utilized Hurst-Kolomogorov framework to evaluate the

performance of climate models (GCM and RCM) rather than to calibrate hydrological models. In their

study, Hurst exponent was used to represent the long-range dependence and evaluate the

reproductivity of variability (Onyutha et al. 2019). However, the benefits of using fractal theory in

model building and calibration have not been discussed. That is, because observed streamflow series

has the inherent feature of fractality, the hydrological model shall be able to reproduce the fractality

of observed data, including self-affinity, periodicity, long-term memory and irregularity.

Since the first hydrological model was developed, proper methods to evaluate the performance of

models have been pursued by hydrology community and a large variety of criteria have been proposed

and used over the years (Pushpalatha et al., 2012). Most of the criteria are based on the squared

residuals or absolute errors (Pushpalatha et al., 2012). Krause et al. (2005) compared nine efficiency

criteria including correlation coefficient ($r^2$), Nash-Sutcliffe efficiency (E), index of agreement (d)

and their variants, but none of them show overall dominance. Kling-Gupta efficiency was developed

by Gupta et al. (2009) and Kling et al. (2012) to provide a diagnostically interesting decomposition of

the Nash-Sutcliffe efficiency, which facilitates the analysis of the relative importance of its different

components (correlation, bias and variability) in the context of hydrological modelling. Apart from

criteria which are used to calculate model errors over the entire test period, there are also many criteria

which focus on a certain period of interests. For example, criteria mentioned above are calculated over

flood periods (Liu et al., 2017; Liu et al., 2019) or dry periods (Demirel et al., 2013). There are also

studies which calibrated hydrological models over different hydrological components other than

streamflow, such as evapotranspiration (Pan et al., 2017), soil moisture (Gao et al., 2015), snow water

equivalent and even glacier melt (Liu et al., 2019). Another approach to modify the performance of





models is hydrological signature (Shafii et al., 2015). Nonetheless, uncertainties of hydrological

signature simulation are often large (Westerberg and McMillan, 2015). Studies also used methods of

loss function in model calibration. For example, Hao et al. (2013) proposed a method based on entropy

theory for constructing the bivariate distribution of drought duration and severity with different

marginal distribution forms. Pechlivanidis et al. (2014) combined conditioned entropy difference

metric and Kling-Gupta efficiency for multi-objective calibration of hydrologic models. Li et al. (2010)

used the Bayesian method for uncertainty assessment of hydrological model estimation.

Since the fractal dimension describes the fractality of streamflow series and two different series may

have the same fractal dimension, the fractal dimension could not be used to calibrate hydrological

model independently. Multi-objective optimization approaches are widely used by hydrological

community (Harlin, 1991; Yapo et al., 1998; Liu et al., 2017; Liu et al., 2019; Pan et al., 2017; Shafii

et al., 2015; Ye et al., 2014). This set the stage of using some uncomprehensive but effective criteria

as targets, such as aforementioned hydrological signatures (Shafii et al., 2015; Westerberg and

McMillan, 2015), statistical targets and fractal criteria.

In this study, a new criterion defined as ratio of fractal dimension (RD) is introduced, as well as a

calibration strategy. The criterion and calibration strategy should be able to consider the self-affinity,

periodicity, long-term memory and irregularity of hydrograph during model calibration. Three

catchments with different climate and geography are used as a case studies. The aim of this study is

to examine the applicability of using RD as one of the targets of multi-objective calibration and

explore the performance of hydrological models when RD is considered. Section 2 describes

differences between RD and classical criteria and how RD is used in calibration (E-RD strategy).

Section 3 contains the brief information of study areas and methods used in this study to investigate

the advantages of RD. Section 4 provides the results and Section 5 provides the discussion. Section 6

is the summary and conclusion. In this study, our goal is to answer the following questions: (1) Is RD

a proper criterion for hydrological modelling, even if the reflection of RD is not as direct as classical

criteria? (2) Could E-RD strategy explicitly improve the performance of hydrological models? (3)

Why can RD be used to improve calibration?

## 2 Ratio of fractal dimension as a criterion of hydrological model calibration

### 2.1 Existing criteria and their drawbacks

Chiew and McMahon (1993) classified calibration criteria into statistical parameters and
dimensionless coefficients. Statistical parameters include mean value, standard deviation, coefficient
of skewness, coefficient of variance and quantile points etc. Dimensionless coefficients include
Pearson correlation coefficient ($r^2$), Nash-Sutcliffe efficiency coefficient (E) and Kling-Gupta
efficiency coefficient etc. Most of the coefficients are based on the squared residuals (Pushpalatha et
al., 2012). According to squared-residuals-based coefficients' calculation formula, approaching of
every simulated individual data to observed data makes the coefficients better. However, this doesn't
make models more convincing if the approaching of simulated individual data is against the physical
feature of models and catchment.

Another deficiency of existing criteria is the preference of particular parts of hydrograph. For example,
statistical parameters are easily influenced by extreme individuals and are with large uncertainties
(Westerberg and McMillan, 2015). Coefficients provide a measure of the overall agreement between
simulation and observation, but are still significantly influenced by particular parts of hydrograph.
High flows make a significant contribution to the value of E and Kling-Gupta efficiency coefficient
(Pushpalatha et al., 2012). Nevertheless, literatures report an underestimation of peak flow when using
E as indicator alone (Jain and Sudheer, 2008).

Overall, there is still large vacancy for calibration criteria which can give consideration to individual
data and whole hydrograph.

### 2.2 Fractal dimensions of time series

Fractality was firstly proposed by Hurst in 1951 to represent some regular patterns of the internal
features of hydrological time series, especially self-affinity and periodicity. Self-affinity is a feature
of a fractal whose pieces are scaled by different amounts in the x- and y-directions, and fractal
dimensions represent the self-affinity of time series. Following the fractal theory, descriptions of
fractality include the Hurst exponent (rescaled range analysis) (Hurst, 1951), Hausdorff dimension
(box-counting dimension or local dimension) (Jelinek et al., 2008; Kenneth, 1990), and correlation





dimension (Grassberger and Procaccia, 1983) etc. The difference of these dimensions is the calculation

scheme of fractal dimensions, and they are numerically related and of dependent significances

theoretically. While the Hurst exponent calculated with rescaled range analysis was more widely used,

the Hausdorff dimension could be expanded to multifractal analysis easily and has perspective

applications in hydrology (Bai et al., 2019; Zhou et al., 2014). For this reason, the Hausdorff

dimension calculated with box-counting method is adopted in this study.

Generally, the Hausdorff dimension of streamflow series represents the magnitude of fluctuation, i.e.,

the fluctuations in river flow are large for large river flow and small for small river flow (Movahed

and Hermanis, 2008). Such feature is also called as long-term correlation, which can be described with

the Hausdorff dimension (Onyutha et al., 2019). Unlike typical statistical evaluation of fluctuation

(such as standard deviation and distribution function), the Hausdorff dimension takes the order of data

into account. Therefore, on the basis of classical criterion controlling water budget, the Hausdorff

dimension can offer useful insight into mechanisms controlling the extreme hydrological events

(including floods, droughts and low flows) (Radziejewski and Kundzewicz, 1997). Another difference

between fractal dimension and classical criteria is the influence of individual (or a small number of)

data. While approaching of every simulated individual data to observed data makes the coefficient

better, it may make the Hausdorff dimension of simulated data closer or farther away from that of

observed data. Given all that, the Hausdorff dimension is proposed in this study in hydrological model

calibration. A good hydrological model shall be able to reproduce the streamflow well in all aspects,

which means reproduced streamflow series and observed streamflow series have similar Hausdorff

dimensions.

The value of Hausdorff dimension of the same time series may be different for different δ. The

difference usually implicates that self-affinity of the time series changes as the resolution changes and

in hydrology specifically, dominant driver of hydrological processes changes. Hausdorff dimension

helps reveal the dominant drivers of hydrological process (Bai et al., 2019). According to this idea,

Hausdorff dimension determines whether the streamflow components are reasonably simulated. In

this study, the largest temporal resolution is set as 365 days (1 year), to leave the inter-annual drivers

out. It is believed that the range of resolution is enough for Hausdorff dimension to reflect drivers of



hydrological processes.

## 2.3 Ratio of fractal dimensions

The box-counting method used to calculate Hausdorff dimension is based on the idea of separating

data into boxes and count the number of boxes (Mandelbrot, 2004). When adopted to analysis of time

series, the box-counting method sums adjacent data up (put adjacent individuals into boxes) and

compares the treated data of various resolutions (different sizes of boxes). Fig. 1 graphically shows

how the box-counting method works with time series. Fig. 1 (a) shows how the number of boxes

needed to cover all data (N) changes when the size of boxes changes (resolution, $\delta$). Fig. 1 (b) shows

the log-linear relationship between N and $\delta$. The definition of Hausdorff dimension D is:

$$D = \frac{\log (N)}{\log (1/\delta)}, \qquad\qquad (1)$$

Where $\delta$ is the size of boxes and N is the number of boxes (Evertsz and Mandelbrot, 1992).

Fig. 1. Flow chart of using box-counting method to calculate the Hausdorff dimension of time series.


As stated before, the observed and simulated streamflow series shall have the same Hausdorff

dimension. In this study, a new criterion named as ratio of dimension (RD) is defined as follow:

$$RD = \frac{D_s}{D_o}, \qquad\qquad (2)$$

where $D_s$ is the Hausdorff dimension of simulated streamflow series and $D_o$ is the Hausdorff

dimension of observed streamflow series. The range of RD is from 0 to $+\infty$. When $RD=1$, the

simulated streamflow series has the same Hausdorff dimension with that of observed streamflow

series, which means that the model is the best in terms of fractals. Nonetheless, the strategy to use

Hausdorff dimension to calibrate hydrological models has not been studied. The relevant examination

of models' performance under the supervision of RD has not been studied, either.

### 2.4 *E-RD* strategy

Obviously, RD, as a metric of self-affinity deviation of simulated streamflow series from observed

series, is not a criterion capable of evaluating the performance of hydrological models by itself. An





immediate thought is to combine RD and another statistical criterion in model calibration.

Three features are demanded for the statistical criterion to be combined with RD. Firstly, the statistical criterion shall be able to evaluate the performance of models in terms of water balance to some extent. Secondly, the statistical criterion shall evaluate the response of streamflow to meteorological forcing. Thirdly, the criterion shall calculate model errors over the entire test period. These features make sure that the strategy meets basic needs. An additional requirement for the statistical criterion used in this study is the popularity of this criterion within the hydrological community. In this manner, the

advantages of RD emerge as well as the benefits of multi-objective calibration based on RD.

Nash-Sutcliffe efficiency coefficient (E), a commonly used criterion since initially proposed (Jain and Sudheer, 2008; Nash and Sutcliffe, 1970), is selected in this study. The calculation of E is:

$$E = 1 - \frac{\Sigma(Q_o - Q_s)^2}{\Sigma(Q_o - \overline{Q_o})^2}, \qquad (3)$$

Where $Q_s$ is the simulated flow, $Q_o$ is the observed flow, $\overline{Q_o}$ is the mean value of the observed

flow.

A set of experiments is processed to illustrate the benefits of using the proposed E-RD strategy to evaluate models. Descriptions of experiments are included in Section 3. Fig. 2 shows the whole process of E-RD strategy.

Fig. 2. Flow chart of E-RD strategy.

## 3 Study area and methodology

### 3.1 Study area

A small catchment located in Tibet named Dong, a medium sized catchment located in southeastern China named Jinhua and a large catchment located in the middle reach of Yangtze River named Xiang

are used in this study.

Dong is a small tributary of the Yarlung Zangbo River, with elevation ranging from 3512 to 5869 m. The area of Dong catchment is about 43.6 km2. The average annual precipitation of study period is 413.5 mm. The average temperature is 10.6 ºC. The high elevation of Dong catchment results in cold climates. Former study has consolidated that snow pack and frozen soil significantly effect

hydrological processes in the Dong catchment (Bai et al., 2019). Meteorological forcing data and

streamflow observation of Dong catchment used in this study are from 2011 to 2014.

Jinhua River is a 5536-km2 catchment of Zhejiang Province, southeastern China. The study area is

subject to Asian monsoon climate, and precipitation is strongly summer-dominant, occurring mostly

from May to September. Based on meteorological data of 42 years (from 1965 to 2006), the mean

annual precipitation in Jinhua catchment is 1847.4 mm. The average temperature is 17.6 ºC. Former

studies show that precipitation data and streamflow data of Jinhua catchment are well matched (Pan

et al., 2018). Meteorological forcing data and streamflow observation of Jinhua catchment used in this

study are from 1965 to 2006.

Xiang River is one of the largest tributaries of the Yangtze River, which flows into the Dongting Lake,

the second largest freshwater lake in mid-China. The area of Xiang catchment is about 82,400 km2

and data of nine meteorological stations are used in this study. Dominated by subtropical monsoon

climate, the mean annual rainfall of the basin ranges from 1400 to 1700 mm and the average annual

temperature is around 17 ºC. The basin experiences floods and droughts frequently, and rainfall is

distributed evenly throughout the year, most of it falling in April to June. According to literatures,

precipitation is the most vital driver for Xiang River (Zhu et al., 2019). Meteorological forcing data

and streamflow observation of Xiang catchment used in this study are from 1987 to 2013.

Fig. 3 shows the topography of all study areas.

Fig. 3. DEM of study areas.


### 3.2 HBV model

The HBV model is a conceptual rainfall-runoff model originally developed by Swedish

Meteorological and Hydrological Institute (SMHI) (Bergström, 1976; Bergström, 1992; Lindström et

al., 1997). The HBV model has been successfully used in many cases (Seibert and Vis, 2012; Tian et

al., 2015; Tian et al., 2016). The HBV model is composed of precipitation and snow accumulation

routines, a soil moisture routine, a quick runoff routine, a baseflow routine and a transform function.

The HBV model takes into account the effect of snow melting and accumulation, which is significant





in the Dong catchment. In this study, the HBV model utilized is provided by Tian et al. (2015), compiled with MATLAB.

The HBV model has 14 parameters to control all above-mentioned hydrological processes. A simple description of seven parameters mentioned later is given here.

The runoff generation routine (response function) of HBV model transforms excess water from the soil moisture zone to runoff. The function consists of one upper, non-linear reservoir and one lower, linear reservoir, presenting fast flow and baseflow separately (Lindström et al., 1997). The fast flow

function has two parameters, namely fast flow factor (KF) and fast flow exponent ($\alpha$). The baseflow (slow flow) function has one parameter, namely baseflow factor (KS). The functions are:

$$Q_f = KF \times UZ^{1+\alpha}, \tag{4}$$

$$Q_b = KS \times LZ, \tag{5}$$

$$Q_T = Q_f + Q_b, \tag{6}$$

where UZ is the water storage of upper reservoir (mm), LZ is the water storage of lower storage (mm), $Q_f$ is the discharge of fast flow (mm), $Q_b$ is the discharge of baseflow (mm) and $Q_T$ is the discharge of total flow (mm).

The replenishment of upper reservoir (effective precipitation) is controlled by field capacity (FC) and effective precipitation exponent ($\beta$) as well as soil moisture and precipitation. The functions are:

$$P_e = \left(\frac{SM}{FC}\right)^{\beta} \times P, \tag{7}$$

where $P_e$ is effective precipitation (mm), SM is soil moisture (mm), P is precipitation (mm).

Percolation is one of the processes which affects replenishment of lower reservoir. In HBV, percolation is represented by a conceptual soil parameter, which represents the maximum amount of replenishment ground water (lower reservoir) from the soil moisture. Obviously, percolation has

significant effect on baseflow and peak of flood. Generally, higher percolation results in higher baseflow and lower peak flow, and vice versa (Abebe et al., 2010). Percolation and baseflow make up the balance of lower reservoir in the HBV model:

$$\Delta_{LZ} = \min(UZ, Percolation) - Q_b, \tag{8}$$

Besides, the HBV model uses capillary transport to simulate the water delivery from soil moisture to

upper reservoir, uses temperature threshold to determine the fraction of snowfall and uses a degree-

day factor to calculate snowmelt.

Other parameters which are not sensitive to RD (see Section 4.2), are not introduced here.

### 3.3 Multi-objective genetic algorithm

A controlled, elitist genetic algorithm (a variant of NSGA-II) (Deb, 2001) is applied in model

calibration. A controlled elitist GA favors individuals with better fitness value (rank) as well as

individuals that can help increase the diversity of the population even if they have a lower fitness

value. An important behavior of this genetic algorithm is, the individual with the best performance

according to anyone of the criterion would be retained with the lowest rank. This makes sure that with

multi-objective genetic algorithm, parameter set with the best possible E could be found and the

following comparison between RD-E and E is reasonable. In this study, $|1 - RD|$ is used as one of

the criteria.

Since HBV has 14 parameters to calibrate, the number of generations is 2800. Each generation has

600 population. The crossover fraction is set as 0.8 (meaning). The Pareto fraction is set as 0.2

(meaning). The population migrates every 20 generations, and the migration fraction is set as 0.5.

These settings make sure that population won't trap in local optimum, which is important because RD

varies in a wider range than traditional criteria. Most of these numbers are the default settings, which

is applicable to most of the questions. Only the number of the population of each generation (600) is

larger than default (200) for finer presentation of Pareto front of the optimization.

All 600 Pareto-optimized solutions of the last generation are used in the following analysis. GA

optimization with the E-RD calibration strategy will not drop population with perfect RD (=1) and

unsatisfactory E. Several representative selected parameter sets and correspondent simulated

streamflow series are deeply studied.

### 3.4 Approach for model evaluation

To investigate the RD's effects in hydrological model evaluation, several tools are utilized.

Pearson's correlation coefficient $r^2$, percentage bias (bias), auto-correlation of observation, auto-

correlation of simulation, relative variance, maximum monthly flow and minimum monthly flow are

used for a comprehensive comparison between models based on RD and traditional hydrological

criteria (E). The best RD model and best E model (typical models) are selected from the last generation of GA calibration for detailed analysis.

To understand how the model is adjusted when RD is used as one of the objectives, the relationship between parameters and RD is analyzed. The distance correlation $r_d^2$ is used to determine whether the variations of model's parameters and RD are related. The relationship between parameters and RD may not be linear, which brings the necessity of using a nonlinear analysis approach rather than Pearson's linear correlation coefficient. Distance correlation is also more robust to data outliers, than

rank correlations.

    To look into the influence on simulation of specific parts of hydrological hydrographs brought by RD, fast flow and baseflow are analyzed separately. The HBV model is slightly modified to output simulated fast flow and baseflow at every time step. Observed streamflow series are divided into the fast and baseflow using the Water Engineering Time Series PROcessing tool (WETSPRO tool)

introduced by William (2009). The E and $r^2$ of simulated fast flow/baseflow to observed fast flow/baseflow are calculated. Hydrographs of the first three years after warming up are shown to visually illustrate the influence of RD on fast flow and baseflow simulation.

## 4 Results and discussion

### 4.1 Overall evaluation of models on the Pareto front

Fig. 4 shows the RD-E relationship of last population of multi-objective calibration in three catchments separately. The ranges of RD of final generation in three cases are different, so as the ranges of E/bias. The ranges of E are 0.60 to 0.69 (Dong), 0.95 to 0.953 (Jinhua) and 0.818 to 0.822 (Xiang). For all cases, the non-significant variation of E indicates that for all selected parameter sets, the E criteria could not fully distinguish them. On the contrary, the ranges of RD are about 0.72 to

$1+2.8 \times 10^{-12}$ (Dong), 0.86 to 1.04 (Jinhua) and 0.85 to 1.01 (Xiang). According to literatures, the biggest difference of Hausdorff dimension of data of the same type is smaller than 0.25 (Hurst, 1951; Rubalcaba, 1997; Meseguer-Ruiz et al., 2019), which indicates the ranges of RD aforementioned are significant. Because that $RD$=1 means that the model is best in terms of fractals (see Section 2.2), the models whose RD is larger than 1 need discussion, too. Noting that the largest RD is very close to



best RD (=1), the largest RD model should be similar with the best RD model. And the GA algorithm

discards most of the models with RD>1 because they are not on the Pareto front. The bias doesn't

change much when RD changes. A tiny difference within 3% occurs for the last generations of Dong,

Jinhua and Xiang. More importantly, change of bias with the change of RD is different for three cases.

For Dong catchment, the bias is firstly getting worse then getting better as RD approaching 1. Besides,

a trend of bias of getting worse for Jinhua and a trend of bias of getting better for Xiang as RD

approaching 1 can be observed. Without more cases, the trend of bias in this study is regarded to be

random. In addition, in Xiang case, there is a break in Fig. 4. On two sides of the break, E is close by

and RD is significantly different.

A single-objective calibration was operated to support the assumptions made in Section 3.3 that, in

this study, the NSGA II algorithm could find the best E. The comparison between results of single-

objective calibration and multi-objective calibration (E-RD strategy) is listed in Table 1.

Fig. 4 E-RD of last generation of GA calibration.

Table 1. Comparison of best E between single-objective calibration and multi-objective calibration

(E-RD strategy).

The models with the best RD, best E and largest RD are selected as typical examples. Fig. 5 is the

simulated streamflow of three examples and observed streamflow as well. For each case, discharge

within a three-year period is shown. Examples with the largest RD are used to verify if the model with

the same RD with observation is the best. Apparently, the simulated hydrographs of typical models in

each case are similar, which agrees with the E and RD in Fig. 4.

Fig. 5. Typical examples with best RD, best E and largest RD (representative three-year hydrograph).


A precondition of adopting the RD-E strategy is the irrelevancy or weak correlation between two

criteria. This precondition could be simply verified by looking into their calculation schemes or by




examining during the multi-objective calibration. The best E and worst E are close according to the result of multi-objective calibration. In this study, the former is apparent and the latter is conducted

and briefly shown in Fig. 4. Figs. 4 and 5 further implicate that only little decrease of E happens when pursuing better RD. In this study, the equifinality of using only E emerges.

Table 2 lists the E values of typical models selected by the E-RD calibration strategy and optimized model. Table 2 confirms the assumption that, in this study, directly analyzing the models calibrated by E-RD calibration strategy is reasonable and efficient. Hydrological signatures including relative

variance, lag-1 auto-correlation, percentage bias and maximum/minimum monthly flow are used to show the effect of RD.

Table 2. Hydrological signatures of typical models in all three cases.

Table 2 shows the hydrological signatures of observed and simulated flow series in three cases. Most hydrological signatures, including lag-1 auto-correlation, relative variation and maximum monthly flow, of simulated series are close by. Lag-1 auto-correlations of simulated series are close with auto-correlations of observed flow series. The lag-1 auto-correlations of series of flow series in Dong case and Xiang case are more than 0.9 while the values in Jinhua case are between 0.75 to 0.77. The relative

variances of flow series in Dong case and Xiang case are smaller than 1, while the values in Jinhua case are more than 1.8. These show the feature of catchments of different types which are well simulated by the HBV model. Maximum and minimum monthly flows of simulated and observed series are significantly different. In all three cases, maximum monthly flows of simulated series are close to each other and slightly smaller than maximum monthly flows of observed flow series.

Minimum monthly flow is the only hydrological signature used in this study that distinguishes the typical models with the best RD and with the best E. In all three cases, the minimum monthly flow of simulated series with the best E is significantly smaller than of the minimum monthly flow observed series. On the contrary, the minimum monthly flow of simulated series with the best RD is close to the minimum monthly flow of observed series in Jinhua and Xiang cases. The minimum monthly flow

of simulated series with the largest RD is worse than that of simulated series with the best RD. In

summary, hydrological signatures illustrate that major effects of RD are on the model's low flow simulation. Therefore, in later sections, low flow related analysis will be more emphasized.

### 4.2 Effect of *RD* on model parameters

All parameter sets in the Pareto frontier of three cases vary. The distance correlations ($r_d^2$) of parameters and RD are used to determine whether the change of parameters is stable. In addition, high value of $r_d^2$ indicates the significant relationship between Hausdorff dimension and these parameters. In this study, a value 0.8 of $r_d^2$ is used as the threshold of being determinative. In this study, the relation between GA-selected parameter sets and E is not shown because E and RD in the Pareto front are highly related and the variance of E is small (see Fig. 4).

Table 3 lists the determinative parameters of three cases respectively. The parameters with $r_d^2 < 0.8$ in all cases are not listed in Table 3. The parameter effective precipitation exponent ($\beta$) and degree-day factor are also listed in Table 3. Effective precipitation exponent is listed in Table 3 because of two reasons: 1) the range of $r_d^2$ of $\beta$ in Jinhua case is from 0.709 to 0.739 in Xiang case, which is better than all unlisted parameters; 2) $\beta$, as well as determinative parameters $\alpha$, KF, KS, is a runoff-

generation-related parameter. The degree-day factor is listed in Table 3 because of two reasons: 1) distance correlation between the degree-day factor and RD is close to 0.8 in Xiang case; 2) in Dong case, distance correlation analysis does not show the significance of ablation of snow to hydrograph. Capillary transport is not determinative parameters to RD in Dong and Jinhua and therefore no further discussion of them is given afterwards. The $r_d^2$ of $\beta$ (0.512) in Dong catchment is smaller than those

in other cases. Fig. 6 shows the relationship between $\beta$ and RD.

Table 3. Determinative parameters and distance correlations ($r_d^2$) between parameters and RD.

\*: $r_d^2 \geq 0.8$


Fig. 6. $\beta$ and RD relationship in three cases.

An explicit relationship between parameters and criteria confirms that the effect of RD is not random.

Six parameters ($\beta$, $\alpha$, fast flow factor, baseflow factor, percolation, degree-day factor) are selected by

distance correlation analysis for further discussion.

The $r_d^2$ of $\alpha$ and RD is larger than 0.740 in all cases. Fig. 7 shows the relationship between $\alpha$ and

RD. The fast flow factor (KF) is related to RD in all cases. Fig. 8 shows the relationship between KF

and RD. The varying patterns of $\alpha$ and KF are the same in three cases. The fast flow exponent $\alpha$

increases when RD approaches to 1 and KF decreases when RD approaches to 1.


Fig. 7. $\alpha$ and RD relationship in three cases.

Fig. 8. KF and RD relationship in three cases.

Fig. 9 shows how fast flow changes with different surface water storages under different KF and $\alpha$

of example models with best RD and best E. For all cases, E selects higher KF and lower $\alpha$ . In

Dong case, the relative difference of fast flow generation between best RD model and best E model is

always around 36%. The difference of fast flow between best RD model and best E model is significant

in Dong case for the whole simulated period. In Jinhua case, the relative difference between the best

RD model and best E model decreases from more than 20% to less than 5%. In Xiang case, the relative

difference between the best RD model and best E model decreases from about 16% to 8%. The

difference between the best RD model and best E model is important during the dry period and reduces

as water storage of upper reservoir increases (the wet period). The relative difference is greater in

Jinhua case than in Xiang case in low flow period but smaller in Jinhua case in high flow period. That

difference between the best E model and best RD model will finally lead to the greater variation of

fast flow in low flow period than in high flow period. There are break points in Xiang case (Fig. 6, 7

and 8), but no evident effects shown in Fig. 5 and 9.

Fig. 9. Response of fast flow to surface water storage. For each case, fast flow responses of typical

models with best RD and E are presented.





The baseflow (slow flow) factor (KS) is related to RD in all cases. Fig. 10 shows the relationship

between KS and RD. The varying patterns of KS are the same in three cases. However, the variation

ranges of KS in three cases are different. The largest value of KS in Dong (0.153) is much larger than

that in Jinhua (0.063) and in Xiang (0.048). The smallest value of KS in Dong (0.016) is also larger

than that in Jinhua (0.005) and in Xiang (0.010).

Fig. 10. KS and RD relationship in three cases.

The percolation is significantly related to RD in all cases. The range of percolation in Dong case is

larger than the others. Fig. 11 shows the relationship between percolation and RD in three cases.

Percolation increases in Dong case and decreases in other two cases when RD increases. The range of

percolation in Dong is larger than in the others. KS and percolation determines the way the HBV

models baseflow.


Fig. 11. Percolation and RD relationship in three cases.

The degree-day factor is significantly related to RD in Xiang case. However, the relationship between

the degree-day factor and RD in Dong and Jinhua is weak. Fig. 12 shows the relationship between the

degree-day factor and RD. The degree-day factor of most selected models of Dong case is smaller

than 0.05, indicating that these models barely have any snow-melt runoff. When RD>0.9, several

models have degree-day factors larger than 7. When RD is around 1, the range of degree-day factor is

8.18 to 11.76, indicating that RD somehow detects the snow-melt runoff in the hydrograph and makes

the HBV model simulate the snow-melt runoff more reasonably. Notably, the RD-selected degree-day

factor in Dong case is too large according to the guidance of HBV (1.5 to 4 mm/day, in Sweden)

(Seibert, 2005), which may result from the unsuitable lumped model structure of HBV in rugged

mountainous catchment.

The degree-day factor in all selected models of Jinhua is large, but the temperature in Jinhua is too

high to have snow accumulation. The distance correlation between the degree-day factor and RD is





weak in Dong and Jinhua case. The range of degree-day factor of most models in Xiang case is from

2.8 to 3.4. The range is small and so as the difference of snow-melt runoff of selected models. Besides,

in Xiang case, only 61 of 9862 days have minus temperature. That is, although the $r_d^2$ of degree-day

factor and RD in Xiang case is large, the snow-melt runoff in Xiang catchment is not influential.

Fig. 12. Degree-day factor and RD relationship in three cases.

As illustrated in Fig. 7, 8, 10 and 11, the three runoff-generation-routine parameters, namely baseflow

factor (KS), fast flow factor (KF) and fast flow exponent ($\alpha$), have the same change patterns in three

cases, suggesting a consistent preference of RD in all cases. Fig. 9 shows the visual difference of fast

flow caused by introducing RD. However, other parameters have different change patterns along with

RD because of different features of catchments (Fig. 6 and 12). For example, the soil parameter $\beta$, as

illustrated by Equation (6), redistributes the precipitation and divides it into effective precipitation and

infiltration. $\beta$ in Dong case and Xiang case increases and $\beta$ in Jinhua decreases when RD is getting

better.

**4.3 Analysis of separated streamflow**

Separated simulated and observed streamflow series further reveal how RD influences model

calibration results. Fig. 13 shows the correlation coefficients between simulated and observed fast

flow/baseflow ($r_f^2$ and $r_b^2$) and Nash-Sutcliffe efficiency coefficient ($E_f$ and $E_b$) of all population

of last generation in three cases and their variation with RD. The observed fast flow and baseflow are

separated from observed total flow using WETSPRO (William, 2009) (see Section 3.4). In Dong case,

both $r_b^2$ and $r_f^2$ slightly decrease as RD approaching 1. However, the range of $r_f^2$ in Dong case is

from 0.02 to 0.15 and the range of $r_b^2$ is from about 0.3 to 0.6, which means no correlation exists

between simulated and observed fast flow and baseflow. The $E_f$ and $E_b$ in Dong case improve to

0.06 and 0.24 respectively. In Jinhua and Xiang cases, all models of last generation of GA simulate

fast flow well. The $r_f^2$ and $E_f$ value in Jinhua case are above 0.95 and 0.94. The $r_f^2$ and $E_f$ in

Xiang case are above 0.78 and 0.70. Surprisingly, there is still an evident improvement of fast flow

simulation due to the application of RD. The major improvement is the performance in baseflow

simulation. The values of criteria of baseflow simulation ($r_b^2$ and $E_b$) are improved from poor to

satisfactory. In Jinhua case, $r_b^2$ is improved from less than 0.1 to more than 0.45 and $E_b$ is improved

from -10 to about 0.38. In Xiang case, $r_b^2$ is improved from about 0.4 to 0.75 and $E_b$ is improved

from -6 to 0.51.

Fig. 13. Correlation coefficients and Nash-Sutcliffe efficiency coefficients on fast flow and baseflow

respectively of models of last generation in three cases.


Fig. 14 and 15 show separated streamflow of typical models and observed streamflow to make a

visible comparison of models based on E and models based on RD. Fig. 14 shows the fast flow and

Fig. 15 shows the baseflow.

Fig. 14. Fast flow of typical models and observations (representative three-years hydrograph).

Fig. 15. Baseflow of typical models and observations (representative three-years hydrograph).

The fast flow response of the best RD model in Dong case matches well to observed fast flow. The

recession of fast flow of the best RD model in Dong case is too fast and the stable value is nearly zero,

which is in the contrast of observation. The fast flow response of best E model in Dong is late, the

recession of fast flow is too slow and fast flow at recession periods is too much. The fast flow response

of largest RD model in Dong is also late, but the fast flow recession is more reasonable. In Jinhua and

Xiang cases, simulated fast flow of all typical models well matches the observation. In all cases, the

fast flow of best RD model is smaller than that of best E model and the difference is greater in low

flow periods, which is consistent with Fig. 9.

In all three cases, the best RD models simulated baseflow well. RD selected models accurately

simulated the seasonal flow variation of three catchments. The amplitude of baseflow fluctuation is

close to separated observation by WETSPRO. The discharge also fits separated observation well. In

all three cases, best E models don't simulate the baseflow well enough. The models with the largest

RD in three cases have different performance. Dong-largest RD model, of which $r_b^2 = 0.82$ and $E_b = 0.25$, is not satisfactory. On the contrary, $r_b^2$ and $E_b$ of the best E model in Dong are 0.87 and 0.79 respectively. The best E models in Jinhua and Xiang case, however, are close to the best RD models. According to Fig. 10 and 11, in Jinhua and Xiang cases, smaller KS and percolation (of best

RD model) make smaller recharge and outflow (baseflow) of lower reservoir and smaller fluctuation of baseflow. In Dong case, bigger percolation increases the recharge and total baseflow and smaller KS extends the period of baseflow recession, making simulated baseflow more consistent with observation (Fig. 15).

Two reasons exist for the unsatisfactory simulation of fast flow in Dong case. The first one is that the

HBV model is not capable to accurately simulate mountainous catchment with snowpack and little gauge data are available for the Dong catchment. The second one is that WETSPRO may fail to correctly separate the short streamflow series of Dong catchment. This needs to be further verified.

Another visual demonstrator of the preference of RD is Fig. 14 and 15. Fast flow generation based on RD is more immediate while baseflow generation based on RD is smoother. Both of them are visually

better than that when RD is not taken into account.

Above results reveal the benefits of using RD and the slight decrease of E. The selection principle based on multi-objective calibration is therefore suggested following two steps: 1) sieving out all parameter sets whose RD is around 1 (in this case, considering the data precision of MATLAB, RD=1); 2) Choosing the parameter set with best E among the sets in Step 1. It's determined that the E-RD

strategy using this selection principle improves the reliability of streamflow components simulation. That is, RD selects responsive fast flow (confirmed in Fig. 14) and smooth baseflow (confirmed in Fig. 15) in all cases.

**5 Conclusion**

This study targeted at examining the possibility of using fractal theory to improve the performance of

hydrological models. The definition of ratio of fractal dimension (RD) was proposed and used as a fractal criterion (against traditional statistical criteria). A scheme which combined RD and Nash-Sutcliffe efficiency coefficient (E) to calibrate hydrological models was developed and examined.

Three study cases named Dong, Jinhua and Xiang were included in the examination. This is the first time (to our best knowledge) that fractal theory was applied to calibrate hydrological models.

Here are the main conclusions of this study:

1) The varying patterns of parameters of runoff generation routine (namely fast flow factor, fast flow exponent and baseflow factor) are similar in all cases of our study.

2) Several parameters were found related to RD in specific cases as a result of specific characteristics of study areas. For instance, E-RD strategy selected the degree-day factors with relatively high value

in Dong case, which is not seen when only E was considered.

3. The E-RD strategy is innovative in hydrological modelling. That is, the E-RD calibration strategy is a potential way to take the fractality of observed streamflow series into model calibration. Since fractal (also regarded as self-affinity) widely exists in nature and fractal of streamflow series is substantial, the RD as a criterion can be a good supplement for hydrological model calibration.

It's noted that the E-RD strategy introduced here needs more case studies to corroborate its capability further. The combination of other traditional statistical criteria and RD shall also be examined. More studies are also needed to dig out more benefits of applying fractal theory in hydrological modelling.

**Acknowledgement**

This study is financially supported by National Key Research and Development Plan "Inter-
governmental Cooperation in International Scientific and Technological Innovation" (2016YFE0122100), the Natural Science Foundation of Zhejiang, China (LZ20E090001) and National Natural Science Foundation of China (91547106). The authors are indebted to PowerChina Huadong Engineering Corporation Limited, who provided meteorological and hydrological data in Dong catchment. National Climate Center of China Meteorological Administration is greatly
acknowledged for providing meteorological data in the Jinhua and Xiang catchments, and Zhejiang Hydrological Bureau is acknowledged for providing hydrological data in the Jinhua River.



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



**Table 1: Comparison of best E between single-objective calibration and multi-objective calibration (E-RD strategy).**

|        | Single-objective (E) | Multi-objective (E) |
| ------ | -------------------- | ------------------- |
| Dong   | 0.696                | 0.690               |
| Jinhua | 0.951                | 0.953               |
| Xiang  | 0.820                | 0.822               |




**Table 2: Hydrological signatures of typical models in all three cases.**

|  |  | Observation | Best RD | Best E | Largest RD |
|---|---|---|---|---|---|
| Auto correlation | Dong | 0.97 | 0.99 | 1.00 | 1.00 |
|  | Jinhua | 0.76 | 0.76 | 0.76 | 0.75 |
|  | Xiang | 0.94 | 0.95 | 0.94 | 0.94 |
| Relative variance | Dong | 0.53 | 0.56 | 0.58 | 0.57 |
|  | Jinhua | 1.87 | 1.87 | 1.87 | 1.89 |
|  | Xiang | 0.99 | 0.82 | 0.92 | 0.92 |
| Maximum monthly flow ($m^3/s$) | Dong | 1.54 | 1.40 | 1.42 | 1.39 |
|  | Jinhua | 531.19 | 497.40 | 503.68 | 496.77 |
|  | Xiang | 4210.01 | 3956.24 | 4027.68 | 4042.94 |
| Minimum monthly flow ($m^3/s$) | Dong | 0.44 | 0.30 | 0.27 | 0.26 |
|  | Jinhua | 60.64 | 58.85 | 50.45 | 60.19 |
|  | Xiang | 961.00 | 975.07 | 812.02 | 840.02 |



**Table 3: Determinative parameters and distance correlations ($r_d^2$) between parameters and RD. *: $r_d^2 \geq 0.8$**

| | $r_d^2$ (range of parameter) | | |
|---|---|---|---|
| | Dong | Jinhua | Xiang |
| Effective precipitation exponent ($\beta$) | 0.363 (0.010~0.012) | 0.709 (0.791~0.911) | 0.739 (0.435~0.499) |
| Fast flow exponent ($\alpha$) | 0.383 (0.100~0.124) | *0.808 (0.473~0.579) | 0.734 (0.677~0.819) |
| Fast flow factor (KF) | *0.853 (0.002~0.005) | *0.812 (0.031~0.056) | *0.823 (0.003~0.006) |
| Baseflow factor (KS) | *0.932 (0.016~0.153) | *0.922 (0.005~0.063) | *0.950 (0.010~0.048) |
| Percolation | *0.879 (1.37~7.00) | *0.841 (1.10~2.34) | *0.959 (1.62~3.16) |
| Capillary transport | 0.122 (0~0.035) | 0.084 (3.84~4.00) | *0.914 (1.91~2.70) |
| Degree-day factor | 0.117 (0.01~12.2) | 0.171 (14.5~15.6) | *0.791 (2.80~4.20) |




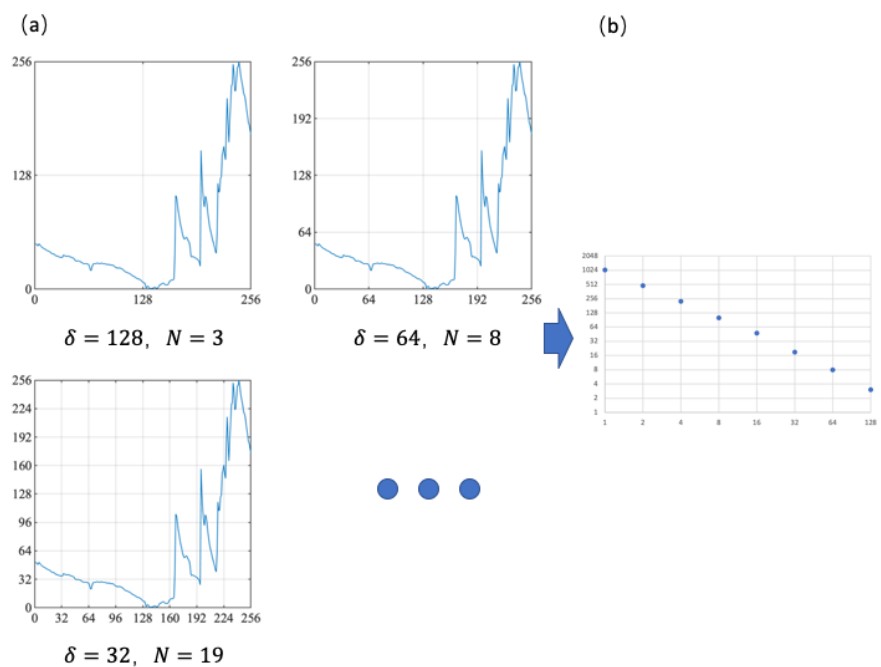

Figure 1: Flow chart of using box-counting method to calculate the Hausdorff dimension of time series.





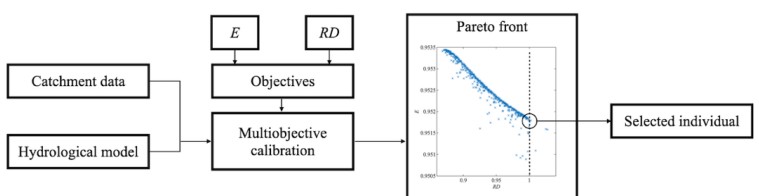

**Figure 2: Flow chart of E-RD strategy.**





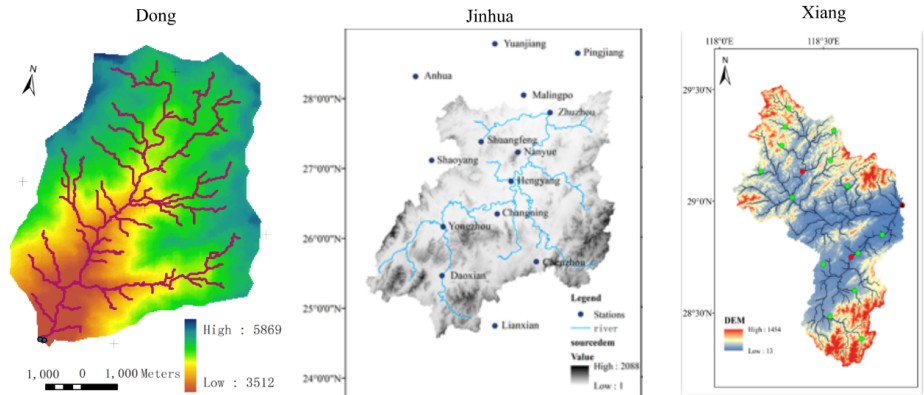

**Figure 3: DEM of study areas.**




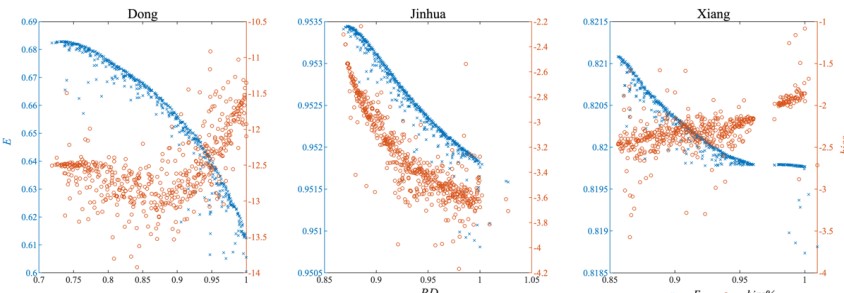

**Figure 4:E-RD of last generation of GA calibration.**




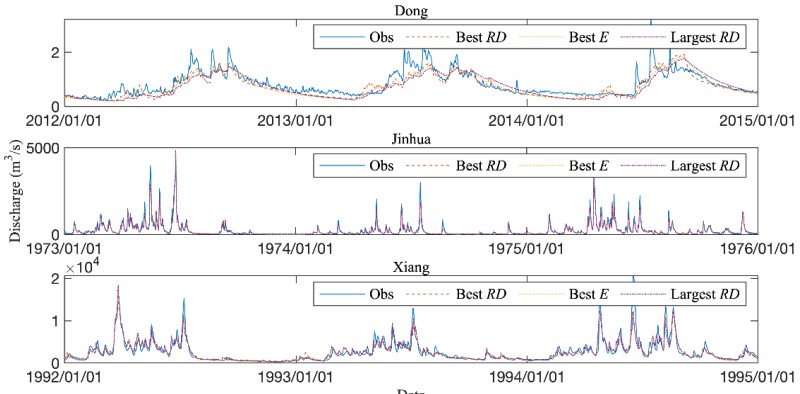

**Figure 5: Typical examples with best RD, best E and largest RD (representative three-year hydrograph).**






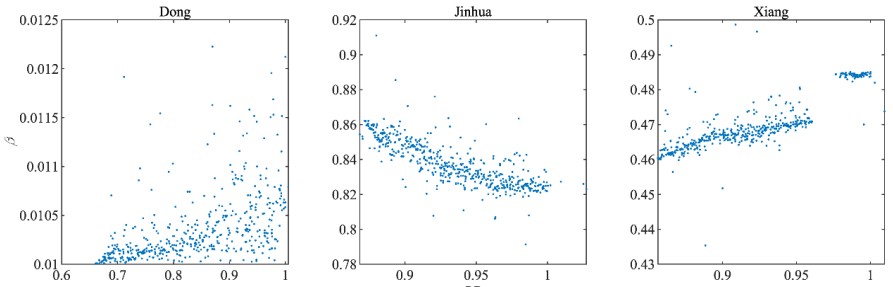

**Figure 6:** $\beta$ and RD relationship in three cases.



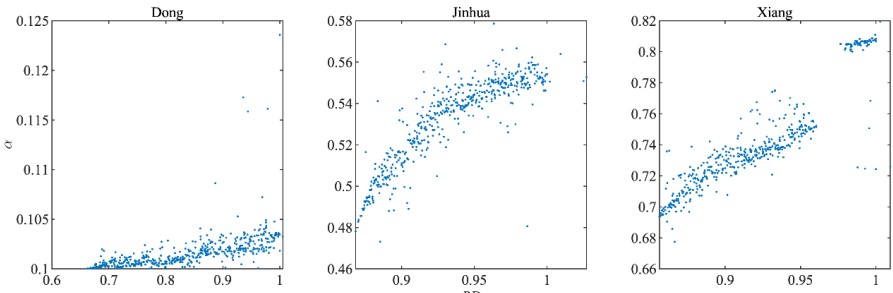

**Figure 7:** $\alpha$ **and RD relationship in three cases.**




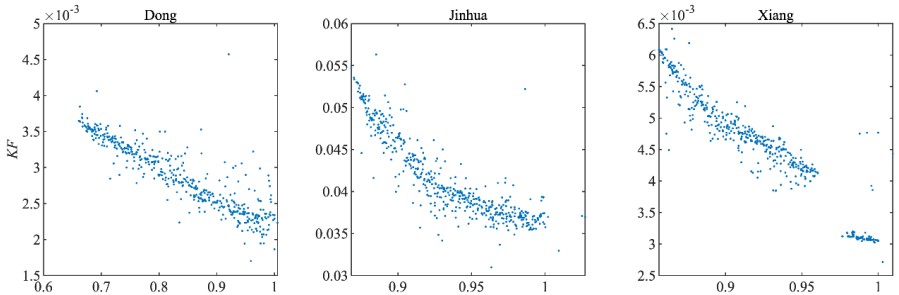

**Figure 8: KF and RD relationship in three cases.**






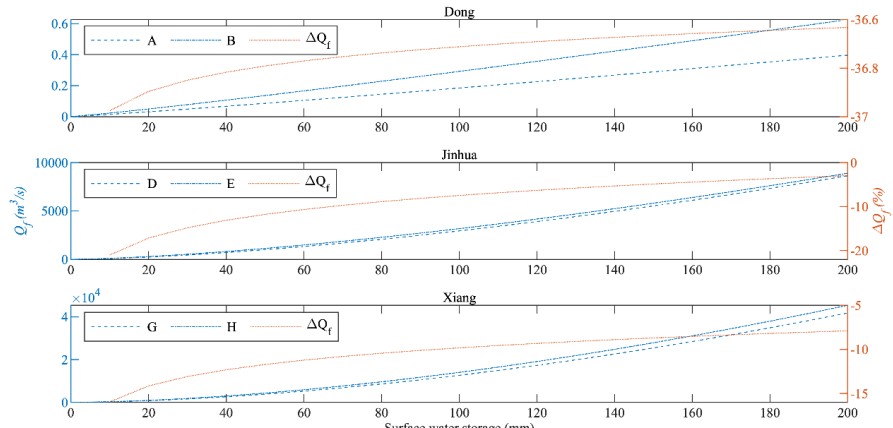

**Figure 9: Response of fast flow to surface water storage. For each case, fast flow responses of typical models with best RD and E are presented.**





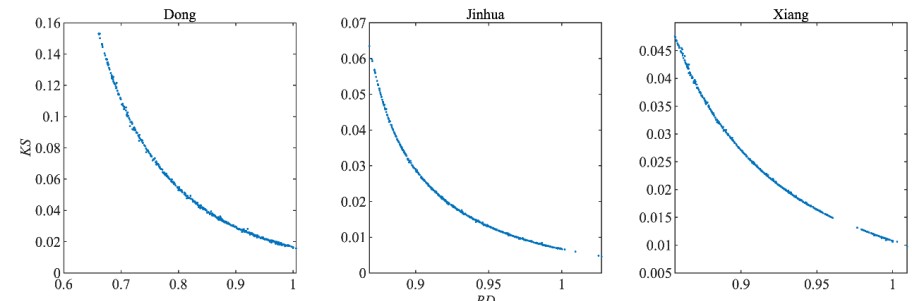


**Figure 10: KS and RD relationship in three cases.**





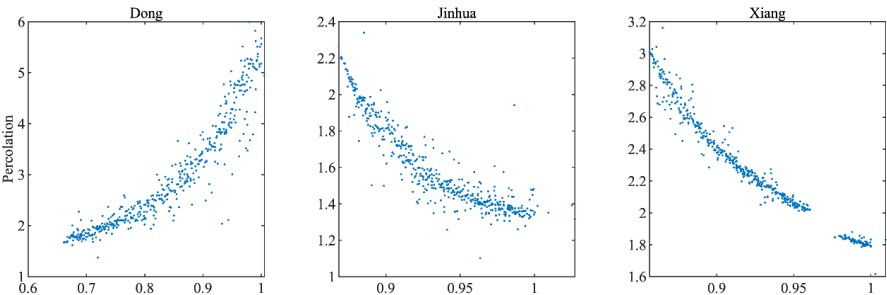

**Figure 11: Percolation and RD relationship in three cases.**






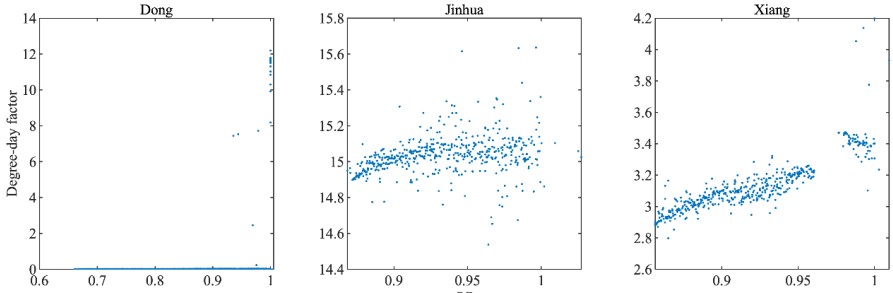

**Figure 12 : Degree-day factor and RD relationship in three cases.**





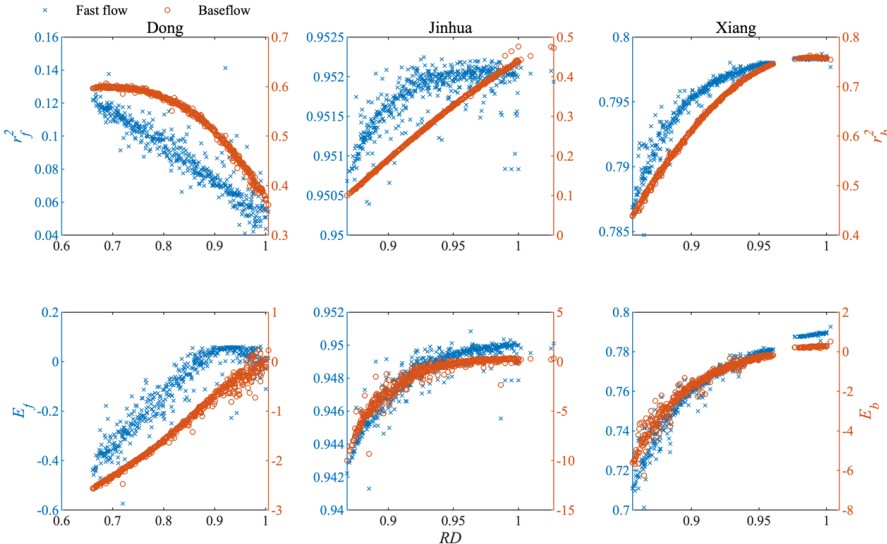

**Figure 13: Correlation coefficients and Nash-Sutcliffe efficiency coefficients on fast flow and baseflow respectively of models of last generation in three cases.**





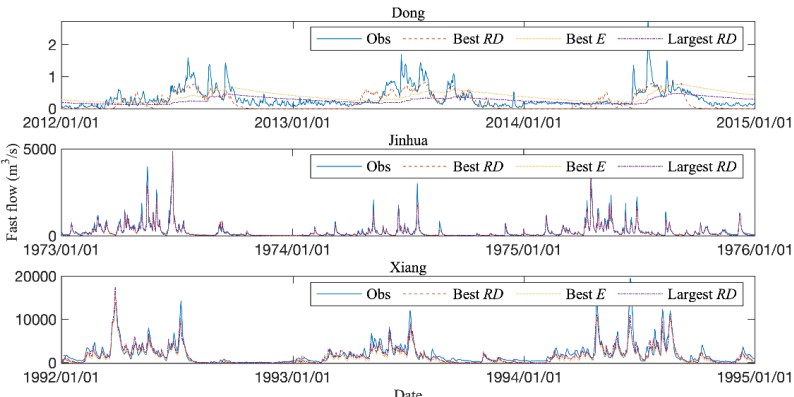

**Figure 14: Fast flow of typical models and observations (representative three-years hydrograph).**






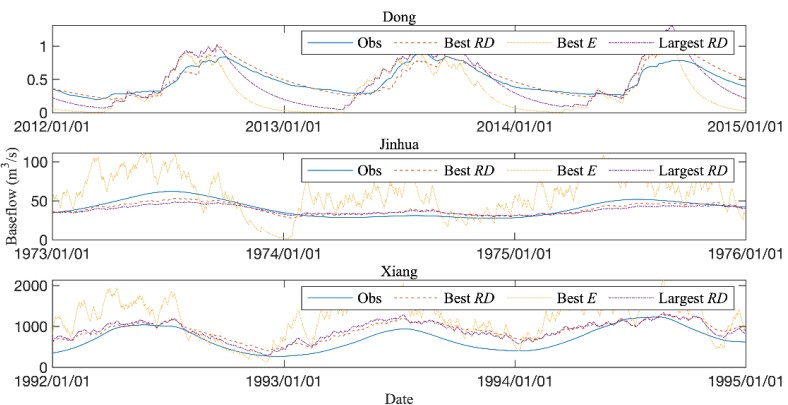

Fig. 15. Baseflow of typical models and observations (representative three-years hydrograph).