# Peer review of "A new fractal-theory-based criterion for hydrological model calibration"

_Hydrology and Earth System Sciences, 2020_

## Referee Comment (RC1) · Anonymous Referee #1 · 9 Dec 2020

GENERAL

The authors proposed a calibration strategy in terms of Hausdorff dimension and Nash-Sutcliffe Efficiency (E) (Nash and Sutcliffe, 1970). The concept that the fractal dimension of observed streamflow should be comparable with that of the model output is an important consideration for hydrological modelling. In Equation (2) of the discussion paper, the authors suggested the metric RD as the ratio of Ds to Do (where Do and Ds denote the Hausdorff dimension of observed and simulated flow, respectively). The calibration strategy the authors are proposing is in terms of E-RD. The authors applied the E-RD strategy to calibrate a selected rainfall-runoff model in applied to a number of catchments in China. In its present form, the manuscript has a number of major areas of concern especially with the use of the proposed E-RD strategy.

[Figure]

COMMENT 1

The metric E which the authors are using in their strategy is known to have a number of issues in its application for assessing "goodness-of-fits". Eventually, the need to modify E has been on the radar of hydrologist for decades. In other words, several variants of E exist to address the issues related to the use and interpretation of the original version from Nash and Sutcliffe (1970) which is still widely applied in hydrology. The question to answer is: why did the authors adopt the original version of E but not any of the existing variants?

COMMENT 2

RD varies from zero to positive infinity (see line 155 of the discussion paper). However, E varies from negative infinity to zero. The point is that both E and RD are relative error measures. For relative error measure, we focus on the "standard" range in which values vary from zero and one with association to imperfect and perfect model, respectively. Therefore, how can a modeler interpret E and RD in a combined way yet the range of the values from each of these metrics is wider than the "standard" one?

COMMENT 3

There is a possibility in modelling that the larger the number of calibration runs, the better the value of the objective function (especially if the parameter spaces are not small). However, the modeler needs to compute both E and RD in each calibration run as a requirement for the strategy being introduced. Thus, application of the introduced strategy brings about the problem of computation time required to reach optimum during calibration of a hydrological model. How can this problem be addressed to ensure application of the introduced strategy is not at the expense of calibration time (especially if the modeler is making use of long-term hydrological series)?

COMMENT 4

The best RD does not guarantee that E will be at its highest value. Furthermore, E
reduces as the modeler searches for the best RD (see lines 330-331 of the discussion paper). This brings about (i) the issue of subjectivity in determining which values of E and RD should be used to select the set of optimal model parameters, (ii) the complication in dealing with the trade-off regarding the decision on which study objective should be preferred to others. To explain (ii), the authors need to note that a modeler may be aiming at reproducing extreme hydrological extremes especially peak high flows, and low flows. Applying the E-RD strategy means, the modeler should also aim at ensuring Ds and Do are the same or very close to one another. The question that the authors need to answer is: How can a modeler deal with the issues (i) and (ii) in application of the calibration strategy being introduced?

COMMENT 5

Sub-flows' separation procedure adopted for this study (incorporated in the tool named WESTPRO) makes use of a number of parameters. The authors never mentioned any values of such parameter in their discussion paper. Examples of such parameters (among others) include recession constants, and the filter parameter. At least two parameter values are required to extract base flow. Again, not less than two parameters are required to filter interflow. Thus, for each river flow time series one requires not less than four parameters to obtain the various sub-flows. The problem is that the choice of this parameters can be largely subjective (even if one takes into account his or her expert judgment in deciding on the parameter values to use for sub-flow filtering of a given streamflow). Moreover, sets of parameters required to separate sub-flows vary from one catchment to another. Finally, there are several methods available for separation of flows (what we also call the baseflow separation techniques). All these problems compound the challenge of using E-RD to judge model performance (or select which calibration run is the best). Furthermore, the overall problems that the authors need to take into account, here, are with respect to the uncertainty (i) due to the choice of the baseflow separation technique (whether manual approach as the authors adopted or automated technique), (ii) the subjectivity of selecting which

parameter values to use in filtering the observed and modeled streamflow. Here, the fact that the same set of parameter values are required to be applied to both observed and modeled streamflows should be considered basic and they need to go beyond it in addressing this comment. Finally, given the above background on sub-flow separation and unanswered question, statements made by the authors in the manuscript citing that the use of RD improves simulations of sub-flows remain claims (or are vague) unless they prove otherwise.

COMMENT 6

The authors attempted to relate optimal values of the model parameters to obtained RD's. In a number of cases (see, for instance, lines 436 and 505) the authors pointed that the selected model lacked capacity to simulate certain hydrological processes. The question to answer is: Why did the authors not take into account the uncertainty in their results due model selection? Models differ with respect to their structures (or underlying assumption and equations). It becomes imperative that the authors need to select at least two models and apply them to various catchments. In doing so, I suggest the authors focus on clear objectives of modeling so as to allow them comprehensively judge the influence of application of RD on the model results. Such objectives may include reproducing (i) extreme peak high flows, (ii) low flows, (iii) fractality in the observed streamflow. Furthermore, results on comparison of RD with model parameter should be put as supplementary material (if they cannot be discarded from the manuscript).

COMMENT 7

Instead of only selecting catchments from China, the authors need to take into account the influence from the differences in climatic conditions on the use of the E-RD strategy. This is because the difficulty in reproducing fractality in observed streamflow from catchments selected across various climatic regions may not be comparable. Furthermore, to guard against manipulations of model inputs, the catchments should be

selected in such a way that their datasets for modeling should be from sources which readers can easily access. There are a number of catchments with complete information such as, hydro-meteorological data, which can be used for rainfall-runoff modelling. To mention, but one example, is the Rainfall-Runoff Library data which can be obtained via https://toolkit.ewater.org.au/Tools/RRL (accessed: 8th December, 2020).

MINOR COMMENTS

Line 9: Change "aims to investigate" to "aims at investigating"

Short forms or words should be removed from the manuscript. A few examples of such words written in short forms include "doesn't" (lines 90,301), "won't" (line 260), and "don't" (line 495).

Line 285: Change "William" to "Willems"

REFERENCE

Nash, J. E., and Sutcliffe, J. V. River flow forecasting through conceptual models part I - a discussion of principles, J. Hydrol., 10, 282–290, doi:10.1016/0022-1694(70)90255-6, 1970.

---

## Referee Comment (RC2) · Anonymous Referee #2 · 5 Jan 2021

General comments

This paper introduces a new calibration criterion based on fractal theory and combines this criterion with a traditional criterion in a multi-objective setting for hydrological model calibration. The hydrological model HBV is applied to three catchments in China with different surface areas and calibrated using the two criteria and the multi-objective calibration algorithm. The additional value of the fractality-based criterion is evaluated in a general way and for different flow components, and relations between the newly-developed criterion and parameters are analysed.

Overall, the paper is reasonable written and presents interesting insights in the use and value of the fractality-based criterion. The authors generally use informative figures to illustrate their results. Several issues need attention such as the structure of the first

three sections, the different lengths of the calibration periods of the three catchments, the analysis of results only for the calibration period (no temporal validation) and the comparison of the observed and simulated flow components. These and other specific comments can be found below. The English spelling, grammar and style can be improved; several examples and other technical corrections can be found below as well.

Specific comments

1. L20-39: The introduction starts with a paragraph about fractality without describing the (hydrological) context. Fractality is used as an additional criterion for model calibration in this paper to obtain calibrated models which perform well for better reasons than when only using traditional criteria based on for instance squared residuals. Hence, a fractality-based criterion is introduced and evaluated as a tool for a more robust model calibration. Therefore, it would be more logical to start the introduction with a description of the pros and cons of existing calibration criteria followed by the introduction of fractal theory as an additional evaluation framework for hydrological models. The last two sentences of sub-section 2.3 (L157-159) typically form (part of) the research gap and are a natural link to the research objective.

2. L23: Terms like 'self-affinity', 'periodicity', 'long-term memory' and 'irregularity' are listed without any explanation. In different contexts these terms might have different meanings. What is the meaning of these terms in this study and which of these terms are quantified by/ included in the ratio of fractal dimensions introduced in this paper?

3. L82: The structure of sections 2 and 3 can be improved. Part of the discussion of the traditional criteria and their cons (and pros) and the fractal dimensions and related indices (sub-sections 2.1 and 2.2) can be included in the introduction (section 1). The description of the fractality-based criterion used in this study (sub-section 2.3) and the calibration strategy (sub-section 2.4) can be merged with section 3. As a consequence, section 2 will disappear.

[Figure]

4. L190-206: The authors use data from three catchments with different sizes and different data periods (and lengths of time series). In particular the data period for the Dong catchment is short (4 years) compared to the other two catchments. What is the influence of these differences in data periods on the results? Does it explain the relatively poor performance of the HBV model for the Dong catchment compared to the other two catchments, particularly for fast and slow flow? Did the authors test their framework with equal data periods for the three catchments (i.e. for Jinhua and Xiang also 4-year time series)? This would be a useful test to isolate the influence of difference lengths of data periods.

5. L211-247: The description of the HBV model is somewhat messy and not complete. For instance, actual evapotranspiration is not described, the order of the fluxes is not logical and the description of the parameters is not consistent with the literature. Since this model has been very frequently used and described in the literature, the authors are advised to reduce the description to a small general paragraph and refer for more details to the literature.

6. L257-263: Although the authors mention that most settings of the calibration algorithm are default ones, it is not completely clear what the meaning of these numbers is and why mostly default settings have been used. Moreover, which 14 HBV parameters need to be calibrated? This is a large number of parameters making the calibration cumbersome. Why not firstly carrying out a sensitivity analysis to select the most dominant parameters?

7. L268: Although the authors compared different observed and simulated signatures (and separated flow components, see next comment), a validation in time and/ or space has not been carried out. It would be very interesting to see how well the HBV model performs for another time period in the three catchments. This would enable a more independent and robust evaluation of the E-RD strategy proposed by the authors.

8. L283-285: The authors compare observed and simulated fast flow and baseflow.

Observed components have been obtained using the WETSPRO tool. However, it is unclear how well the division into streamflow components is done by this tool (also mentioned by the authors in sub-section 4.3). It might well be that observed and simulated components describe a (totally) different flow mechanism. For the Dong catchment, this results in a poor performance for streamflow components and for the other two catchments in a good performance. What is the principle used by WETSPRO to separate flow components and to what extent is that principle related to the concepts of the HBV model? This needs more discussion by the authors.

9. L358: Section 4.2: the selection of parameters for further analysis is not completely clear and straightforward. The authors mention a threshold for the distance correlation (is this a correlation value of a squared correlation value), but they do not consequently apply this threshold. Furthermore, the (unexpected) high correlation between RD and the degree-day factor for the (mainly) rainfed Xiang catchment needs more discussion.

10. L413-416: Would it be possible to relate the different parameter values for different catchments to differences in characteristics of these catchments (e.g. slope, soil types, size)? See also e.g. lines 451-454.

Technical corrections

1. L26 and elsewhere: "studies" instead of "literatures".

2. L26: "indices" instead of 'indexes".

3. L38: ". . . the hydrological model shall be able to"; what is meant by this sentence? How does it relate to the previous sentence?

4. L41: "hydrological" instead of "hydrology".

5. L50: "interest" instead of "interests".

6. L57: What kind of loss function is referred to here?

7. L90-92: What is meant with this sentence? Do you have an example of a situation

where the individual data points are well simulated but physical behaviour of the model (for a particular catchment) is not realistic?

8. L110-111: What do the authors mean with 'dependent significances theoretically'?

9. L120: What is meant with 'classical criterion controlling water budget'? A calibration criterion which compares observed and simulated water balances?

10. L130: What is the meaning of the 'delta' symbol?

11. L132: This part of the sentence is not clear and is an example which needs reformulation.

12. L209: It would be better to use the same colours for the three DEMs.

13. L265: What is the E-RD calibration strategy? Probably here the explanation from sub-section 2.4 can be used.

14. L266: "corresponding" instead of "correspondent".

15. L276: Do you have a reference for 'distance correlation'?

16. L295: "1.0" instead of "1+2.8x10-12".

17. L298: What is the meaning of 'significant' here?

18. L298-299: And when RD is smaller than 1? In this study RD often is lower than 1 and sometime only slightly higher than 1. Why is this the case?

19. L314: Xiang catchment has a different (wrong?) x-axis.

20. L318: Are the best RD, best E and largest RD selected from the Pareto front?

21. L329-330: This sentence is not clear.

22. L359: "front" instead of "frontier".

23. L377-378: The ranges of the parameters in Table 3 need units. In addition, the

ranges for parameter BETA are very small and all below 1. How do these ranges compare with recommended ranges from the literature?

24. L386: The first part of this line is not correct; the correlation'for the Dong catchment is smaller than 0.74.

25. L391-392: Please include a unit for KF in Fig. 8 and switch the axes; i.e. RD is the dependent variable and the parameters are the independent variables. This also applies to other figures with RD as a function of parameter values.

26. L409-410: What is the meaning of the letters A, B, D, E, G and H?

---

## Author Comment (AC1) · 1 Feb 2021

Dear Editors and Reviewers: Thanks for your kind comments about our manuscript. Your comments are not only helpful but also inspiring. The comments provide new perspectives to understand the application of fractal theory in hydrological modeling. We have studied the reviewers' comments carefully and made responses in the following texts. We are looking forward for further advice from you.

Kind regards, Zhixu Bai, Yao Wu, Di Ma, Zixia Wang

Responses to the reviewers' comments:

Reviewer #1:

[Figure]

COMMENT 1 The metric E which the authors are using in their strategy is known to have a number of issues in its application for assessing "goodness-of-fits". Eventually, the need to modify E has been on the radar of hydrologist for decades. In other words, several variants of E exist to address the issues related to the use and interpretation of the original version from Nash and Sutcliffe (1970) which is still widely applied in hydrology. The question to answer is: why did the authors adopt the original version of E but not any of the existing variants?

RESPONSE 1 Thanks for your kind comment. As far as we know, although the different modification versions of E have been studied for decades, there are still no dominant dimensionless coefficients to measure the performance of hydrological models. When only one metric should be used with RD in our study (else the calibration and selection of parameter sets could be too complex to understand the effects of introducing RD), there are not many choices. We finally chose E rather than KGE or other variants of E because the pros and cons of E are more familiar for hydrologists, and this original version is still most often used in hydrological model calibration.

This concept will be included in Section 2.4 E-RD strategy of our final manuscript: "Another reason to choose E in our study is that the pros and cons of E are more familiar for hydrologists than other metrics, and this original version is still mostly often used in hydrological model calibration."

COMMENT 2

RD varies from zero to positive infinity (see line 155 of the discussion paper). However, E varies from negative infinity to zero. The point is that both E and RD are relative error measures. For relative error measure, we focus on the "standard" range in which values vary from zero and one with association to imperfect and perfect model, respectively. Therefore, how can a modeler interpret E and RD in a combined way yet the range of the values from each of these metrics is wider than the "standard" one?

RESPONSE 2

Thanks for your kind comment. RD varies from zero to positive infinity, but the RD value of a perfect model should be equal to 1 because the simulated streamflow series and the observed streamflow series have the same Hausdorff dimensions. We found that a small range of E near the best E in certain cases corresponds to a relatively large range of RD. Besides, there is always a set of parameters makes RD=1 and E close to the best E. Therefore, we applied a genetic algorithm to find individuals with smallest value of objectives. The flow chart is below (Figure 2 in our manuscript).

Figure 2: Flow chart of E-RD strategy. In the multi-objective optimization, we made some adjustments. The objectives used in the multi-objective optimization are 1-E and |1-RD| (see Line 255).

COMMENT 3

There is a possibility in modelling that the larger the number of calibration runs, the better the value of the objective function (especially if the parameter spaces are not small). However, the modeler needs to compute both E and RD in each calibration run as a requirement for the strategy being introduced. Thus, application of the introduced strategy brings about the problem of computation time required to reach optimum during calibration of a hydrological model. How can this problem be addressed to ensure application of the introduced strategy is not at the expense of calibration time (especially if the modeler is making use of long-term hydrological series)?

RESPONSE 3

We made an experiment to show the effects of introducing two objectives into an automatic calibration to the computation time.

We made an experiment to compare the runs needed for finding the best E (single-objective calibration) and the Pareto optimum of E and RD of HBV model used in this study. The calibration algorithm and parameters are the same with those in our original manuscript.

The results show that the multi-objective calibration took 2160 seconds to run 106 generations (63600 individuals) while the single-objective calibration took 1170 seconds to run 51 generations (31150 individuals).

Besides, to overcome the problem of computation time in multi-objective calibration of hydrological models, hydrologists have adopted several different types of methods. In our study, we have adopted NSGA II genetic algorithm and parallel computing technique to accelerate the calibration.

All in all, the introduction of a new strategy will increase the time required, and several methods were adopted. The calibration time has been controlled to a reasonable range in our study. When the E-RD strategy is used with distributed models, more techniques such as parameters' sensitivity analysis could be applied to reduce the number of parameters to be calibrated.

COMMENT 4

The best RD does not guarantee that E will be at its highest value. Furthermore, E reduces as the modeler searches for the best RD (see lines 330-331 of the discussion paper). This brings about (i) the issue of subjectivity in determining which values of E and RD should be used to select the set of optimal model parameters, (ii) the complication in dealing with the trade-off regarding the decision on which study objective should be preferred to others. To explain (ii), the authors need to note that a modeler may be aiming at reproducing extreme hydrological extremes especially peak high flows, and low flows. Applying the E-RD strategy means, the modeler should also aim at ensuring Ds and Do are the same or very close to one another. The question that the authors need to answer is: How can a modeler deal with the issues (i) and (ii) in application of the calibration strategy being introduced?

RESPONSE 4

Thanks for your kind comment. We would like to respond from two aspects. Firstly, in

multi-objective calibration, the objectives, generally, cannot be at their highest values at the same time. And if they can, the introduction of multiple objectives becomes worthless because a single-objective calibration is able to achieve the same results. Secondly, we believe that RD could help modelers find the best fractality of simulated series. The improved performance (of low flows in our study) is the by-products of the improvement of fractality. We believe the issue (ii) proposed by the reviewer is not a drawback of our strategy because our strategy improves the simulation of low flows and has little effects on high flows. In other words, our strategy provides a better metric. Based on above, our answer is: a modeler may make gentle adjustments of our strategy to make it more suitable for his/her own cases. But the introduction of RD, by making the Hausdorff dimensions of simulated series and observed series closer, could improve the performance and the internal rationality (components of streamflow in our study) of hydrological models.

COMMENT 5

Sub-flows' separation procedure adopted for this study (incorporated in the tool named WESTPRO) makes use of a number of parameters. The authors never mentioned any values of such parameter in their discussion paper. Examples of such parameters (among others) include recession constants, and the filter parameter. At least two parameter values are required to extract base flow. Again, not less than two parameters are required to filter interflow. Thus, for each river flow time series one requires not less than four parameters to obtain the various sub-flows. The problem is that the choice of this parameters can be largely subjective (even if one takes into account his or her expert judgment in deciding on the parameter values to use for sub-flow filtering of a given streamflow). Moreover, sets of parameters required to separate subflows vary from one catchment to another. Finally, there are several methods available for separation of flows (what we also call the baseflow separation techniques). All these problems compound the challenge of using E-RD to judge model performance (or select which calibration run is the best). Furthermore, the overall problems that the authors need to

take into account, here, are with respect to the uncertainty (i) due to the choice of the baseflow separation technique (whether manual approach as the authors adopted or automated technique), (ii) the subjectivity of selecting which parameter values to use in filtering the observed and modeled streamflow. Here, the fact that the same set of parameter values are required to be applied to both observed and modeled streamflows should be considered basic and they need to go beyond it in addressing this comment. Finally, given the above background on sub-flow separation and unanswered question, statements made by the authors in the manuscript citing that the use of RD improves simulations of sub-flows remain claims (or are vague) unless they prove otherwise.

RESPONSE 5

Thanks for your kind comment. We put the parameters of WETSPRO here. And we are pleased to provide the value of parameters into the revised manuscript. Notably, the WETSPRO tool could separate the streamflow into fast flow and slow flow first, and then separate the fast flow into overland flow and interflow. In our study, only the first step is applied and only the first-step-related parameters of WETSPRO are listed in the table below. We selected the parameters by following the procedure. In WETSPRO's procedure, the parameters are selected one by one. For each parameter/step, there is a corresponding criterion. Thus, the separated streamflow components are relatively objective. Fig. R1-5 is an example of the objective procedure of selection. In this step, the user selects the w-parameter filter, which represents the case-specific average fraction of the quick flow volumes over the total flow volumes. According to the literature, the filtered baseflow should be close to the total streamflow in dry periods (Willems, 2009). The selection can be considered relatively objective.

Fig. R1-5 An example of the objective procedure of selection.

Table R1-5 Parameters of WETSPRO

The description of Table R1-5 is as follows: "Table R1-5 lists the parameters of WETSPRO in three cases. The recession constants are close to each other. The wparameter filter, representing the case-specific average fraction of the quick flow volumes over the total flow volumes, shows the difference. The w-parameter filter of Dong catchment is 0.14, smaller than the other catchments, meaning that less proportion of total flow in Dong is baseflow, showing the catchment features of small area and high slope."

COMMENT 6

The authors attempted to relate optimal values of the model parameters to obtained RD's. In a number of cases (see, for instance, lines 436 and 505) the authors pointed that the selected model lacked capacity to simulate certain hydrological processes. The question to answer is: Why did the authors not take into account the uncertainty in their results due model selection? Models differ with respect to their structures (or underlying assumption and equations). It becomes imperative that the authors need to select at least two models and apply them to various catchments. In doing so, I suggest the authors focus on clear objectives of modeling so as to allow them comprehensively judge the influence of application of RD on the model results. Such objectives may include reproducing (i) extreme peak high flows, (ii) low flows, (iii) fractality in the observed streamflow. Furthermore, results on comparison of RD with model parameter should be put as supplementary material (if they cannot be discarded from the manuscript).

RESPONSE 6

Thanks for your kind comment. In our study, we analyzed the parameters' behavior when RD is taken into consideration instead. Besides, we trust that our performance of models is good enough for our cases. We would like to add the analysis about the objectives suggested by the reviewer. We agree that more objectives could make our study of E-RD strategy more comprehensive. Table 2 now becomes:

"The high-flow percentiles ($Q_5$) and low-flow percentiles ($Q_{75}$) are reasonable in three cases for all typical models. However, the high-flow percentiles and low-flow

percentiles of best RD models are still closest to the observation."

COMMENT 7

Instead of only selecting catchments from China, the authors need to take into account the influence from the differences in climatic conditions on the use of the E-RD strategy. This is because the difficulty in reproducing fractality in observed streamflow from catchments selected across various climatic regions may not be comparable. Furthermore, to guard against manipulations of model inputs, the catchments should be selected in such a way that their datasets for modeling should be from sources which readers can easily access. There are a number of catchments with complete information such as, hydro-meteorological data, which can be used for rainfall-runoff modelling. To mention, but one example, is the Rainfall-Runoff Library data which can be obtained via https://toolkit.ewater.org.au/Tools/RRL (accessed: 8th December, 2020).

RESPONSE 7

Thanks for your kind comment. We agree that the selection of catchments across various climatic regions leads to a more convincing result. However, in our manuscript, the three catchments are located in very different climatic regions (see Section 3.1). Dong is a small catchment with continental plateau climate. Xiang is a large catchment dominated by Dominated by subtropical monsoon climate. Jinhua is subject to Asian monsoon climate and effected by typhoon in summer. And we are glad to use open-source data and models in our following studies.

MINOR COMMENTS

RESPONSE

We'll make the corrections as suggested.

[Figure]

**Fig. 1.** Figure 2

Time series
Filtered baseflow
Slope recession constant baseflow

**Fig. 2.** Fig. R1-5

| Parameter | Dong | Jinhua | Xiang |
|---|---|---|---|
| Recession constant (days) | 90 | 80 | 90 |
| w-parameter filter | 0.14 | 0.43 | 0.38 |

**Fig. 3.** Table R1-5

|                          |        | Observation | Best RD | Best E  | Largest RD |
|--------------------------|--------|-------------|---------|---------|------------|
| Auto correlation         | Dong   | 0.97        | 0.99    | 1.00    | 1.00       |
|                          | Jinhua | 0.76        | 0.76    | 0.76    | 0.75       |
|                          | Xiang  | 0.94        | 0.95    | 0.94    | 0.94       |
| Relative variance        | Dong   | 0.53        | 0.56    | 0.58    | 0.57       |
|                          | Jinhua | 1.87        | 1.87    | 1.87    | 1.89       |
|                          | Xiang  | 0.99        | 0.82    | 0.92    | 0.92       |
| Maximum monthly flow ($m^3/s$) | Dong   | 1.54    | 1.40    | 1.42    | 1.39       |
|                          | Jinhua | 531.19      | 497.40  | 503.68  | 496.77     |
|                          | Xiang  | 4210.01     | 3956.24 | 4027.68 | 4042.94    |
| Minimum monthly flow ($m^3/s$) | Dong   | 0.44    | 0.30    | 0.27    | 0.26       |
|                          | Jinhua | 60.64       | 58.85   | 50.45   | 60.19      |
|                          | Xiang  | 961.00      | 975.07  | 812.02  | 840.02     |
| High flow percentiles ($Q_5$) ($m^3/s$) | Dong   | 1.93    | 1.44    | 1.49    | 1.38       |
|                          | Jinhua | 752.00      | 745.02  | 734.12  | 740.28     |
|                          | Xiang  | 6048.50     | 5817.06 | 5586.92 | 5817.08    |
| low flow percentiles ($Q_{75}$) ($m^3/s$) | Dong   | 0.50    | 0.39    | 0.40    | 0.38       |
|                          | Jinhua | 37.77       | 38.55   | 37.80   | 37.31      |
|                          | Xiang  | 803.75      | 790.95  | 845.76  | 744.52     |

**Fig. 4.** Table 2

---

## Author Comment (AC2) · 1 Feb 2021

Dear Editors and Reviewers:

Thanks for your kind comments about our manuscript. Your comments are not only helpful but also inspiring. The comments provide new perspectives to understand the application of fractal theory in hydrological modeling. We have studied the reviewers' comments carefully and made responses in the following texts. We are looking forward for further advice from you.

Kind regards, Zhixu Bai, Yao Wu, Di Ma, Zixia Wang

Responses to the reviewers' comments:

[Figure]

Reviewer 2#:

Specific comments

1. L20-39: The introduction starts with a paragraph about fractality without describing the (hydrological) context. Fractality is used as an additional criterion for model calibration in this paper to obtain calibrated models which perform well for better reasons than when only using traditional criteria based on for instance squared residuals. Hence, a fractality-based criterion is introduced and evaluated as a tool for a more robust model calibration. Therefore, it would be more logical to start the introduction with a description of the pros and cons of existing calibration criteria followed by the introduction of fractal theory as an additional evaluation framework for hydrological models. The last two sentences of sub-section 2.3 (L157-159) typically form (part of) the research gap and are a natural link to the research objective.

Response:

Thanks for reviewer's kind comment. We will change the sequences of our introduction as suggested in our final manuscript.

2. L23: Terms like 'self-affinity', 'periodicity', 'long-term memory' and 'irregularity' are listed without any explanation. In different contexts these terms might have different meanings. What is the meaning of these terms in this study and which of these terms are quantified by/ included in the ratio of fractal dimensions introduced in this paper?

Response:

Thanks for your kind comment. We are glad to add the explanations about "self-affinity", "periodicity", "long-term memory" and "irregularity" following the revised manuscript: "The self-affinity of time series is the similarity of fine-resolution small parts and coarse-resolution large parts of data. Hausdorff dimension is defined and calculated based on the self-affinity of data series."

"The periodicity and long-term memory of time series referred by its fractality are highly

related. Long-term memory is the feature that the effect of an event in a series may persist for a relatively long time. Long-term memory of hydrological time series is usually studied with rescaled range analysis (Hurst, 1951)."

"The irregularity of a fractal series refers to the unpredictable changes in a time series, which is a feature of chaos system."

3. L82: The structure of sections 2 and 3 can be improved. Part of the discussion of the traditional criteria and their cons (and pros) and the fractal dimensions and related indices (sub-sections 2.1 and 2.2) can be included in the introduction (section 1). The description of the fractality-based criterion used in this study (sub-section 2.3) and the calibration strategy (sub-section 2.4) can be merged with section 3. As a consequence, section 2 will disappear.

Response:

Thanks for your kind comment. We have rearranged our manuscript as the reviewer suggested.

4. L190-206: The authors use data from three catchments with different sizes and different data periods (and lengths of time series). In particular, the data period for the Dong catchment is short (4 years) compared to the other two catchments. What is the influence of these differences in data periods on the results? Does it explain the relatively poor performance of the HBV model for the Dong catchment compared to the other two catchments, particularly for fast and slow flow? Did the authors test their framework with equal data periods for the three catchments (i.e. for Jinhua and Xiang also 4-year time series)? This would be a useful test to isolate the influence of difference lengths of data periods.

Response:

Thanks for your kind comment and advice. We made a comparison of the modeling performance based on 4-year time series, and the results are shown below.

[Figure]

Fig. R2-4 Comparison of the E-RD strategy with 4-year data and whole period in Xiang case.

Fig. R2-4 will not be put in the manuscript. According to Fig. R2-4, the E-RD strategy would not change its behavior with the lengths of data in our study. The relevant description will be:

"To get rid of the possible influence of the lengths of time series, a comparison of the multi-objective calibration with the same length of data is made. The results show that, at least in the cases of this study, the E-RD strategy would not change its behavior with the lengths of data."

We agree that the length of data may limit the performance of HBV in Dong case, but as we showed in manuscript, the E of Dong case is about 0.7, which is good enough for an integrated model.

5. L211-247: The description of the HBV model is somewhat messy and not complete. For instance, actual evapotranspiration is not described, the order of the fluxes is not logical and the description of the parameters is not consistent with the literature. Since this model has been very frequently used and described in the literature, the authors are advised to reduce the description to a small general paragraph and refer for more details to the literature.

Response:

Thanks for your kind comment and advice. Following the reviewer's advice, we have modified the description of HBV model as follows: "The HBV model is a conceptual rainfall-runoff model originally developed by Swedish Meteorological and Hydrological Institute (SMHI) (Bergström, 1976; Bergström, 1992; Lindström et al., 1997). The model has been successfully used in many cases (Seibert and Vis, 2012; Tian et al., 2015; Tian et al., 2016). The HBV model is composed of precipitation and snow accumulation routines, a soil moisture routine, a quick runoff routine, a baseflow routine and

a transform function. The HBV model takes into account the effect of snow melting and accumulation, which is significant in the Dong catchment. The actual evapotranspiration is calculated with a linear function. Two conceptual runoff reservoirs, the upper reservoir and the lower reservoir are included in HBV model."

6. L257-263: Although the authors mention that most settings of the calibration algorithm are default ones, it is not completely clear what the meaning of these numbers is and why mostly default settings have been used. Moreover, which 14 HBV parameters need to be calibrated? This is a large number of parameters making the calibration cumbersome. Why not firstly carrying out a sensitivity analysis to select the most dominant parameters?

Response:

Thanks for the reviewer's kind comment. The settings are determined to make sure the calibration can find the Pareto's optimal. And the default settings, as shown by the results, successfully helped us achieve our goals. We would add the reference of the meaning of the parameters in the revised manuscript:

"The meanings of settings can be found in Deb (2001)."

The 14 calibrated parameters of HBV model will be listed and added to Section 3.2. We didn't select the most dominant parameters for calibration for two reasons: 1. Three catchments used in our study are located in different climatic regions and thus have different dominant processes and corresponding dominant parameters. It is not convenient for parameter comparison if different parameters are used in model calibration 2. As an integrated conceptual model, HBV has the advantage of time saving. For example, in the Xiang case, it costs 2160 seconds to run 106 generations (63600 individuals). It's a durable time cost for the calibration of hydrological models.

7. L268: Although the authors compared different observed and simulated signatures (and separated flow components, see next comment), a validation in time and/ or space

has not been carried out. It would be very interesting to see how well the HBV model performs for another time period in the three catchments. This would enable a more independent and robust evaluation of the E-RD strategy proposed by the authors.

Response:

Thanks for the reviewer's kind comment. Since we have used all data of these three catchments in our study, it's hard to make a validation of time with another time period. One reason is that there is only 4 years' data in the Dong catchment. However, we believe the comparison made with different periods (with the length of 4 years) could help doing this.

The comparison lays in the response to comment 4#.

8. L283-285: The authors compare observed and simulated fast flow and baseflow. Observed components have been obtained using the WETSPRO tool. However, it is unclear how well the division into streamflow components is done by this tool (also mentioned by the authors in sub-section 4.3). It might well be that observed and simulated components describe a (totally) different flow mechanism. For the Dong catchment, this results in a poor performance for streamflow components and for the other two catchments in a good performance. What is the principle used by WETSPRO to separate flow components and to what extent is that principle related to the concepts of the HBV model? This needs more discussion by the authors.

Response:

Thanks for the reviewer's kind comment. We believe that the poor performance for streamflow components for the Dong catchment is determined by the moderate performance for total streamflow for the Dong catchment. However, the metrics ($E$ and $r^2$) of fast and slow flow show evident improved trends when RD increases.

The WETSPRO could separate the streamflow into fast flow and slow flow first, and then separate the fast flow into overland flow and interflow. In our study, only the first

[Figure]

step is applied and only the first-step-related parameters of WETSPRO are listed in the table below. We selected the parameters by following the procedure. In WETSPRO's procedure, the parameters are selected one by one. For each parameter/step, there is a corresponding criterion. Thus, the separated streamflow components are relatively comparatively objective. Fig. R1-5 is an example of the objective procedure of selection. In this step, the user selects the w-parameter filter, which represents the case-specific average fraction of the quick flow volumes over the total flow volumes. According to the literature, the filtered baseflow should be close to the total streamflow in dry periods (Willems, 2009). The selection can be considered relatively objective.

Fig. R2-8 An example of the objective procedure of selection. The methodology uses multiple and non-commensurable measures of information derived from the river flow series by means of a number of sequential time series processing tasks. It's derived from the recursive digital filter(Willems, 2009). To briefly introduce the characteristics of WETSPRO to our readers, we would add the following description in Section 3.4:

"WETSPRO separates fast flow and slow flow on the basis of filter theory, using several filter parameters including recession constant, average fraction of fast flow volumes over the total flow volumes, etc." We applied the WETSPRO tool to observed and simulated streamflow series. In this way, the standard of separating streamflow into components could be the same for observation and simulation. We added the following sentence to explain this in sub-section 4.3:

"The simulated total flow is also separated with the WETSPRO tool to make the principle of separation of simulation and observation same."

9. L358: Section 4.2: the selection of parameters for further analysis is not completely clear and straightforward. The authors mention a threshold for the distance correlation (is this a correlation value of a squared correlation value), but they do not consequently apply this threshold. Furthermore, the (unexpected) high correlation between RD and the degree-day factor for the (mainly) rainfed Xiang catchment needs more discussion.

Response:

Thanks for the reviewer's kind comment.

We decided to replace the sentence "The parameters with r_dˆ2<0.8 in all cases are not listed in Table 3." with "distance correlation (r_dˆ2) is used to illustrate the non-linear relationship between E and RD in the Pareto's optimal."

We will add the description of distance correlation in Section 3.4 as follows:

"The distance correlation, as a multivariate measure of dependence, calculates the correlation of distances between points to means. The distance correlation is believed to have better performance when solving problems with non-linear data or extreme values (Székely, Rizzo and Bakirov, 2007)."

We would add the discussion about the high correlation between RD and the degree-day factor for the (mainly) rainfed Xiang catchment as following:

"By checking the temperature series in the Xiang catchment, we find there are 61 days (out of 27 years) when the average temperature is below 0°C. Actually, since the Xiang catchment is large, there are snow events somewhere in the catchment almost every year. The low temperature may be covered by the averaging, but the E-RD strategy captured it and illustrate this by noticeable value of degree-day factor."

10. L413-416: Would it be possible to relate the different parameter values for different catchments to differences in characteristics of these catchments (e.g. slope, soil types, size)? See also e.g. lines 451-454.

Response:

Thanks for the reviewer's kind comment. We used to focus on the differences between different parameter sets of each catchment. We would add the discussion about the different values of the same parameters in different catchments as follows: "The KS of best-E in the three cases follows the sequence of catchment area. This agrees with the

regular pattern that the concentration time of slow flow is highly related with the area of catchment." (Section 4.2) "The percolation in Dong case is larger than the others, which is the reflection of Dong catchment's arid climate. The percolation in Jinhua case is larger than the percolation in Xiang case, because the slope in Jinhua catchment is larger." (Section 4.2)

Technical corrections:

Response:

Thanks for the reviewer's careful work. We will correct as suggested in our final manuscript.

[Figure]

**Fig. 1.** Fig. R2-4

[Figure]

**Fig. 2.** Fig. R2-8

---

## Author Response (AR2)

Dear Editors and Reviewers:

Thanks for your kind comments about our manuscript. Your comments are very helpful to improve the quality of our manuscript. We have studied the reviewers' comments carefully and made responses in the following texts.

Kind regards,

Zhixu Bai, Yao Wu, Di Ma, Yue-Ping Xu

Responses to the reviewers' comments:

Reviewer #1:

**COMMENT 1**

The metric E which the authors are using in their strategy is known to have a number of issues in its application for assessing "goodness-of-fits". Eventually, the need to modify E has been on the radar of hydrologist for decades. In other words, several variants of E exist to address the issues related to the use and interpretation of the original version from Nash and Sutcliffe (1970) which is still widely applied in hydrology. The question to answer is: why did the authors adopt the original version of E but not any of the existing variants?

**RESPONSE 1**

Thanks for your kind comment.

As far as we know, although the different modification versions of $E$ have been studied for decades, there are still no dominant dimensionless coefficients to measure the performance of hydrological models. When only one metric should be used with $RD$ in our study (else the calibration and selection of parameter sets could be too complex to understand the effects of introducing $RD$), there are not many choices. We finally chose $E$ rather than $KGE$ or other variants of $E$ because the pros and cons of $E$ are more familiar for hydrologists, and this original version is still most often used in hydrological model calibration.

This concept will be included in Section 2.4 *E-RD* strategy of our final manuscript:

"Another reason to choose $E$ in our study is that the pros and cons of $E$ are more familiar for hydrologists than other metrics, and this original version is still mostly often used in hydrological model calibration." (line 157 to 159)

**COMMENT 2**

RD varies from zero to positive infinity (see line 155 of the discussion paper). However, E varies from negative infinity to zero. The point is that both E and RD are relative error measures. For relative error measure, we focus on the "standard" range in which values vary from zero and one with association to imperfect and perfect model, respectively. Therefore, how can a modeler interpret E and RD in a combined way yet the range of the values from each of these metrics is wider than the "standard" one?

*RESPONSE 2*

Thanks for your kind comment. $RD$ varies from zero to positive infinity, but the $RD$ value of a perfect model should be equal to 1 because the simulated streamflow series and the observed streamflow series have the same Hausdorff dimensions.

We found that a small range of $E$ near the best $E$ in certain cases corresponds to a relatively large range of $RD$. Besides, there is always a set of parameters makes $RD = 1$ and $E$ close to the best $E$.

Therefore, we applied a genetic algorithm to find individuals with the smallest value of objectives. The flow chart is shown below (Figure 2 in our manuscript).

[Figure]

**Figure 1: Flow chart of E-RD strategy.**

In the multi-objective optimization, we made some adjustments. The objectives used in the multi-objective optimization are $1 - E$ and $|1 - RD|$.

**COMMENT 3**

There is a possibility in modelling that the larger the number of calibration runs, the better the value of the objective function (especially if the parameter spaces are not small). However, the modeler needs to compute both E and RD in each calibration run as a requirement for the strategy being introduced. Thus, application of the introduced strategy brings about the problem of computation time required to reach optimum during calibration of a hydrological model. How can this problem be addressed to ensure application of the introduced strategy is not at the expense of calibration time (especially if the modeler is making use of long-term hydrological series)?

**RESPONSE 3**

We made an experiment to show the effects of introducing two objectives into an automatic calibration to the computation time.

We made an experiment to compare the runs needed for finding the best $E$ (single-objective calibration) and the Pareto optimum of $E$ and $RD$ of HBV model used in this study. The calibration algorithm and parameters are the same with those in our original manuscript.

The results show that the multi-objective calibration took 2160 seconds to run 106 generations (63600 individuals) while the single-objective calibration took 1170 seconds to run 51 generations (31150 individuals).

Besides, to overcome the problem of computation time in multi-objective calibration of hydrological models, hydrologists have adopted several types of methods. In our study, we have adopted NSGA II genetic algorithm and parallel computing technique to accelerate the calibration.

All in all, the introduction of a new strategy will increase the time required, and several methods were adopted. The calibration time has been controlled to a reasonable range in our study. When the $E$ - $RD$ strategy is used with distributed models, more techniques such as parameters' sensitivity analysis and parallel computing could be applied to reduce the number of parameters to be calibrated.

**COMMENT 4**

The best RD does not guarantee that E will be at its highest value. Furthermore, E reduces as the modeler searches for the best RD (see lines 330-331 of the discussion paper). This brings about (i) the issue of subjectivity in determining which values of E and RD should be used to select the set of optimal model parameters, (ii) the complication in dealing with the trade-off regarding the decision on which study objective should be preferred to others. To explain (ii), the authors need to note that a modeler may be aiming at reproducing extreme hydrological extremes especially peak high flows, and low flows. Applying the E-RD strategy means, the modeler should also aim at ensuring Ds and Do are the same or very close to one another. The question that the authors need to answer is: How can a modeler deal with the issues (i) and (ii) in application of the calibration strategy being introduced?

*RESPONSE 4*

Thanks for your kind comment. We would like to respond from two aspects.

Firstly, in multi-objective calibration, the objectives, generally, cannot be at their highest values at the same time. And if they can, the introduction of multiple objectives becomes worthless because a single-objective calibration is able to achieve the same results.

Secondly, we believe that $RD$ could help modelers find the best fractality of simulated series. The improved performance (of low flows in our study) is the by-products of the improvement of fractality. We believe the issue (ii) proposed by the reviewer is not a drawback of our strategy because our strategy improves the simulation of low flows and has little effects on high flows. In other words, our strategy provides a better metric. Based on above, our answer is: a modeler may make gentle adjustments of our strategy to make it more suitable for his/her own cases. But the introduction of $RD$, by making the Hausdorff dimensions of simulated series and observed series closer, could improve the performance and the internal rationality (components of streamflow in our study) of hydrological models.

COMMENT 5

Sub-flows' separation procedure adopted for this study (incorporated in the tool named

WESTPRO) makes use of a number of parameters. The authors never mentioned any values of such parameter in their discussion paper. Examples of such parameters (among others) include recession constants, and the filter parameter. At least two parameter values are required to extract base flow. Again, not less than two parameters are required to filter interflow. Thus, for each river flow time series one requires not less than four parameters to obtain the various sub-flows. The problem is that the choice of this parameters can be largely subjective (even if one takes into account his or her expert judgment in deciding on the parameter values to use for sub-flow filtering of a given streamflow). Moreover, sets of parameters required to separate subflows vary from one catchment to another. Finally, there are several methods available for separation of flows (what we also call the baseflow separation techniques). All these problems compound the challenge of using E-RD to judge model performance (or select which calibration run is the best). Furthermore, the overall problems that the authors need to take into account, here, are with respect to the uncertainty (i) due to the choice of the baseflow separation technique (whether manual approach as the authors adopted or automated technique), (ii) the subjectivity of selecting which parameter values to use in filtering the observed and modeled streamflow. Here, the fact that the same set of parameter values are required to be applied to both observed and modeled streamflows should be considered basic and they need to go beyond it in addressing this comment. Finally, given the above background on sub-flow separation and unanswered question, statements made by the authors in the manuscript citing that the use of RD improves simulations of sub-flows remain claims (or are vague) unless they prove otherwise.

*RESPONSE 5*

Thanks for your kind comment. We put the parameters of WETSPRO here. And we are pleased to provide the value of parameters into the revised manuscript. Notably, the WETSPRO tool could separate the streamflow into fast flow and slow flow first, and then separate the fast flow into overland flow and interflow. In our study, only the first step was applied and only the first-step-related parameters of WETSPRO are listed in the table below. We selected the parameters by following the procedure shown below. In WETSPRO's procedure, the parameters are selected one by one. For each

parameter/step, there is a corresponding criterion. Thus, the separated streamflow components are relatively objective. Fig. R1-5 is an example of the objective procedure of selection. In this step, the user selects the w-parameter filter, which represents the case-specific average fraction of the quick flow volumes over the total flow volumes. According to the literature, the filtered baseflow should be close to the total streamflow in dry periods (Willems, 2009). The selection can be considered relatively objective.

[Figure]

Fig. R1-5 An example of the objective procedure of selection.

Table 4 Parameters of WETSPRO

| Parameter | Dong | Jinhua | Xiang |
|---|---|---|---|
| Recession constant (days) | 90 | 80 | 90 |
| w-parameter filter | 0.14 | 0.43 | 0.38 |

The description of Table 4 is as follows:

"Table 4 lists the parameters of WETSPRO in three cases. The recession constants are close to each other. The w-parameter filter, representing the case-specific average fraction of the quick flow volumes over the total flow volumes, shows the difference. The w-parameter filter of Dong catchment is 0.14, smaller than the other catchments,

meaning that baseflow occupies less proportion of total flow in Dong, showing the catchment features of small area and high slope." (line 455 to 460)

The authors attempted to relate optimal values of the model parameters to obtained RD's. In a number of cases (see, for instance, lines 436 and 505) the authors pointed that the selected model lacked capacity to simulate certain hydrological processes. The question to answer is: Why did the authors not take into account the uncertainty in their results due model selection? Models differ with respect to their structures (or underlying assumption and equations). It becomes imperative that the authors need to select at least two models and apply them to various catchments. In doing so, I suggest the authors focus on clear objectives of modeling so as to allow them comprehensively judge the influence of application of RD on the model results. Such objectives may include reproducing (i) extreme peak high flows, (ii) low flows, (iii) fractality in the observed streamflow. Furthermore, results on comparison of RD with model parameter should be put as supplementary material (if they cannot be discarded from the manuscript).

*RESPONSE 6*

Thanks for your kind comment. In our study, we analyzed the parameters' behavior when $RD$ is taken into consideration instead. Besides, we trust that our performance of models is good enough for our cases.

We have added the analysis about the objectives suggested by the reviewer in the revised manuscript. We agree that more objectives could make our study of $E$-$RD$ strategy more comprehensive. New Table 2 is now becomes:

| | | Observation | Best $RD$ | Best $E$ | Largest $RD$ |
|---|---|---|---|---|---|
| | Dong | 0.97 | 0.99 | 1.00 | 1.00 |
| Auto correlation | Jinhua | 0.76 | 0.76 | 0.76 | 0.75 |
| | Xiang | 0.94 | 0.95 | 0.94 | 0.94 |
| Relative | Dong | 0.53 | 0.56 | 0.58 | 0.57 |

| | | | | | |
|---|---|---|---|---|---|
| variance | Jinhua | 1.87 | 1.87 | 1.87 | 1.89 |
| | Xiang | 0.99 | 0.82 | 0.92 | 0.92 |
| Maximum | Dong | 1.54 | 1.40 | 1.42 | 1.39 |
| monthly flow | Jinhua | 531.19 | 497.40 | 503.68 | 496.77 |
| $(m^3/s)$ | Xiang | 4210.01 | 3956.24 | 4027.68 | 4042.94 |
| Minimum monthly | Dong | 0.44 | 0.30 | 0.27 | 0.26 |
| flow | Jinhua | 60.64 | 58.85 | 50.45 | 60.19 |
| $(m^3/s)$ | Xiang | 961.00 | 975.07 | 812.02 | 840.02 |
| High flow | Dong | 1.93 | 1.44 | 1.49 | 1.38 |
| percentiles | Jinhua | 752.00 | 745.02 | 734.12 | 740.28 |
| $(Q_5)(m^3/s)$ | Xiang | 6048.50 | 5817.06 | 5586.92 | 5817.08 |
| low flow | Dong | 0.50 | 0.39 | 0.40 | 0.38 |
| percentiles | Jinhua | 37.77 | 38.55 | 37.80 | 37.31 |
| $(Q_{75})(m^3/s)$ | Xiang | 803.75 | 790.95 | 845.76 | 744.52 |

"The high-flow percentiles ($Q_5$) and low-flow percentiles ($Q_{75}$) are reasonable in three cases for all typical models. However, the high-flow percentiles and low-flow percentiles of best $RD$ models are still closest to the observation." (line 343 to 345)

COMMENT 7

Instead of only selecting catchments from China, the authors need to take into account the influence from the differences in climatic conditions on the use of the E-RD strategy. This is because the difficulty in reproducing fractality in observed streamflow from catchments selected across various climatic regions may not be comparable. Furthermore, to guard against manipulations of model inputs, the catchments should be selected in such a way that their datasets for modeling should be from sources which readers can easily access. There are a number of catchments with complete information such as, hydro-meteorological data, which can be used for rainfall-runoff modelling. To mention, but one example, is the Rainfall-Runoff Library data which can be obtained

via https://toolkit.ewater.org.au/Tools/RRL (accessed: 8th December, 2020).

*RESPONSE 7*

Thanks for your kind comment. We agree that the selection of catchments across various climatic regions leads to a more convincing result. However, in our manuscript, the three catchments are located in very different climatic regions (see Section 3.1). Dong is a small catchment with continental plateau climate. Xiang is a large catchment dominated by subtropical monsoon climate. Jinhua is subject to Asian monsoon climate and effected by typhoon in summer.

We are glad to use open-source data and models in our following studies. Thanks for your nice suggestion.

MINOR COMMENTS

*RESPONSE*

We'll make the corrections as suggested.

Dear Editors and Reviewers:

Thanks for your kind comments about our manuscript. Your comments are very helpful to improve the quality of our manuscript. We have studied the reviewers' comments carefully and made responses in the following texts.

Kind regards,

Zhixu Bai, Yao Wu, Di Ma, Yue-Ping Xu

Responses to the reviewers' comments:

Reviewer 2#:

Specific comments

1. L20-39: The introduction starts with a paragraph about fractality without describing the (hydrological) context. Fractality is used as an additional criterion for model calibration in this paper to obtain calibrated models which perform well for better reasons than when only using traditional criteria based on for instance squared residuals. Hence, a fractality-based criterion is introduced and evaluated as a tool for a more robust model calibration. Therefore, it would be more logical to start the introduction with a description of the pros and cons of existing calibration criteria followed by the introduction of fractal theory as an additional evaluation framework for hydrological models. The last two sentences of sub-section 2.3 (L157-159) typically form (part of) the research gap and are a natural link to the research objective.

*Response:*

Thanks for the reviewer's kind comment. We have changed the sequences of our introduction as suggested in our revised manuscript.

2. L23: Terms like 'self-affinity', 'periodicity', 'long-term memory' and 'irregularity' are listed without any explanation. In different contexts these terms might have different meanings. What is the meaning of these terms in this study and which of these terms are quantified by/ included in the ratio of fractal dimensions introduced in this paper?

*Response:*

Thanks for your kind comment. We are glad to add the explanations about "self-affinity", "periodicity", "long-term memory" and "irregularity" in the revised manuscript:

"The self-affinity of time series is the similarity of fine-resolution small parts and coarse-resolution large parts of data. Hausdorff dimension is defined and calculated based on the self-affinity of data series."

"The periodicity and long-term memory of time series referred by its fractality are highly related. Long-term memory is the feature that the effect of an event in a series may persist for a relatively long time. Long-term memory of hydrological time series is usually studied with rescaled range analysis (Hurst, 1951)."

"The irregularity of a fractal series refers to the unpredictable changes in a time series, which is a feature of chaos system."

3. L82: The structure of sections 2 and 3 can be improved. Part of the discussion of the traditional criteria and their cons (and pros) and the fractal dimensions and related indices (sub-sections 2.1 and 2.2) can be included in the introduction (section 1). The description of the fractality-based criterion used in this study (sub-section 2.3) and the calibration strategy (sub-section 2.4) can be merged with section 3. As a consequence, section 2 will disappear.

*Response:*

Thanks for your kind comment. We have rearranged our manuscript as the reviewer suggested.

4. L190-206: The authors use data from three catchments with different sizes and different data periods (and lengths of time series). In particular, the data period for the Dong catchment is short (4 years) compared to the other two catchments. What is the influence of these differences in data periods on the results? Does it explain the relatively poor performance of the HBV model for the Dong catchment compared to the other two catchments, particularly for fast and slow flow? Did the authors test their

framework with equal data periods for the three catchments (i.e. for Jinhua and Xiang also 4-year time series)? This would be a useful test to isolate the influence of difference lengths of data periods.

*Response:*

Thanks for your kind comment and advice.

We made a comparison of the modeling performance based on 4-year time series, and the results are shown below.

[Figure]

Fig. R2-4 Comparison of the $E$-$RD$ strategy with 4-year data and whole period in Xiang case.

Fig. R2-4 will not be put in our revised manuscript. According to Fig. R2-4, the $E$-$RD$ strategy would not change its behavior with the lengths of data in our study. The relevant description will be:

"To get rid of the possible influence of the lengths of time series, a comparison of the multi-objective calibration with the same length of data is made. The results show that, at least in the cases of this study, the $E$-$RD$ strategy would not change its behavior with the lengths of data."

We agree that the length of data may limit the performance of HBV in Dong case, but as we showed in the revised manuscript, the $E$ of Dong case is about 0.7, which is good enough for an integrated model.

5. L211-247: The description of the HBV model is somewhat messy and not complete. For instance, actual evapotranspiration is not described, the order of the fluxes is not logical and the description of the parameters is not consistent with the literature. Since this model has been very frequently used and described in the literature, the authors are

advised to reduce the description to a small general paragraph and refer for more details to the literature.

*Response:*

Thanks for your kind comment and advice. Following the reviewer's advice, we have modified the description of HBV model as follows:

"The HBV model is a conceptual rainfall-runoff model originally developed by Swedish Meteorological and Hydrological Institute (SMHI) (Bergström, 1976; Bergström, 1992; Lindström et al., 1997). The model has been successfully used in many cases (Seibert and Vis, 2012; Tian et al., 2015; Tian et al., 2016). The HBV model is composed of precipitation and snow accumulation routines, a soil moisture routine, a quick runoff routine, a baseflow routine and a transform function. The model takes into account the effect of snow melting and accumulation, which is important in the Dong catchment. The actual evapotranspiration is calculated with a linear function. Two conceptual runoff reservoirs, the upper reservoir and the lower reservoir are included in HBV model."

6. L257-263: Although the authors mention that most settings of the calibration algorithm are default ones, it is not completely clear what the meaning of these numbers is and why mostly default settings have been used. Moreover, which 14 HBV parameters need to be calibrated? This is a large number of parameters making the calibration cumbersome. Why not firstly carrying out a sensitivity analysis to select the most dominant parameters?

*Response:*

Thanks for the reviewer's kind comment. The settings of the calibration algorithm are determined to make sure the calibration can find the Pareto's optimal. And the default settings, as shown by the results, successfully helped us achieve our goals. We have added the reference of the meaning of the parameters in the revised manuscript:

"The meanings of settings can be found in Deb (2001)."

The 14 calibrated parameters of HBV model have been listed and added to Section 3.2 in the revised manuscript.

We didn't select the most dominant parameters for calibration for two reasons:

1) Three catchments used in our study are located in different climatic regions and thus have different dominant processes and corresponding dominant parameters. It is not convenient for parameter comparison if different parameters are used in model calibration

2) As an integrated conceptual model, HBV has the advantage of time saving. For example, in the Xiang case, it costs 2160 seconds to run 106 generations (63600 individuals). It's a durable time cost for the calibration of hydrological models.

7. L268: Although the authors compared different observed and simulated signatures (and separated flow components, see next comment), a validation in time and/ or space has not been carried out. It would be very interesting to see how well the HBV model performs for another time period in the three catchments. This would enable a more independent and robust evaluation of the E-RD strategy proposed by the authors.

*Response:*

Thanks for the reviewer's kind comment. Since we have used all data of these three catchments in our study, it's hard to make a validation of time with another time period. One reason is that there is only 4 years' data in the Dong catchment. However, we believe the comparison made with different periods (with the length of 4 years) could help doing this.

The comparison lays in the response to comment 4#.

8. L283-285: The authors compare observed and simulated fast flow and baseflow. Observed components have been obtained using the WETSPRO tool. However, it is unclear how well the division into streamflow components is done by this tool (also mentioned by the authors in sub-section 4.3). It might well be that observed and simulated components describe a (totally) different flow mechanism. For the Dong catchment, this results in a poor performance for streamflow components and for the other two catchments in a good performance. What is the principle used by WETSPRO to separate flow components and to what extent is that principle related to the concepts

of the HBV model? This needs more discussion by the authors.

*Response:*

Thanks for the reviewer's kind comment.

We believe that the poor performance for streamflow components for the Dong catchment is determined by the moderate performance of total streamflow for the Dong catchment. However, the metrics ($E$ and $r^2$) of fast and slow flow show evidently improved trends when $RD$ increases.

The WETSPRO can separate the streamflow into fast flow and slow flow first, and then separate the fast flow into overland flow and interflow. In our study, only the first step was applied and only the first-step-related parameters of WETSPRO are listed in the table below. We selected the parameters by following the procedure shown below. In WETSPRO's procedure, the parameters are selected one by one. For each parameter/step, there is a corresponding criterion. Thus, the separated streamflow components are relatively comparatively objective. Fig. R1-5 is an example of the objective procedure of selection. In this step, the user selects the w-parameter filter, which represents the case-specific average fraction of the quick flow volumes over the total flow volumes. According to the literature, the filtered baseflow should be close to the total streamflow in dry periods (Willems, 2009). The selection can be considered relatively objective.

[Figure]

Fig. R2-8 An example of the objective procedure of selection.

The methodology uses multiple and non-commensurable measures of information derived from the river flow series by means of a number of sequential time series processing tasks. It's derived from the recursive digital filter(Willems, 2009). To briefly introduce the characteristics of WETSPRO to our readers, we have added the following description in Section 3.4 in the revised manuscript:

"WETSPRO separates fast flow and slow flow on the basis of filter theory, using several filter parameters including recession constant and average fraction of fast flow volumes over the total flow volumes etc."

We applied the WETSPRO tool to observed and simulated streamflow series. In this way, the standard of separating streamflow into components could be the same for observation and simulation. We have added the following sentence to explain this in sub-section 4.3:

"The simulated total flow is also separated with the WETSPRO tool to make the principle of separation of simulation and observation same."

9. L358: Section 4.2: the selection of parameters for further analysis is not completely clear and straightforward. The authors mention a threshold for the distance correlation

(is this a correlation value of a squared correlation value), but they do not consequently apply this threshold. Furthermore, the (unexpected) high correlation between RD and the degree-day factor for the (mainly) rainfed Xiang catchment needs more discussion.

*Response:*

Thanks for the reviewer's kind comment.

We decided to replace the sentence "The parameters with $r_d^2 < 0.8$ in all cases are not listed in Table 3." with "distance correlation ($r_d^2$) is used to illustrate the non-linear relationship between $E$ and $RD$ in the Pareto's optimal."

We have added the description of distance correlation in Section 3.4 as follows:

"The distance correlation, as a multivariate measure of dependence, calculates the correlation of distances between points to means. The distance correlation is believed to have better performance when solving problems with non-linear data or extreme values (Székely, Rizzo and Bakirov, 2007)."

We have added the discussion about the high correlation between RD and the degree-day factor for the (mainly) rainfed Xiang catchment as following (line 435 to 440):

"By checking the temperature series in the Xiang catchment, we find there are 61 days (out of 27 years) when the average temperature is below 0°C. Actually, since the Xiang catchment is large, there are snow events somewhere in the catchment almost every year. The low temperature may be covered by averaging, but the $E$-$RD$ strategy captured it and illustrated this by noticeable value of degree-day factor."

10. L413-416: Would it be possible to relate the different parameter values for different catchments to differences in characteristics of these catchments (e.g. slope, soil types, size)? See also e.g. lines 451-454.

*Response:*

Thanks for the reviewer's kind comment. We have added the discussion about the different values of the same parameters in different catchments as follows:

"The $KS$ of best-$E$ in the three cases follows the sequence of catchment area. This agrees with the regular pattern that the concentration time of slow flow is highly related with the area of catchment." (Section 4.2, line 406 to 408)

"The percolation in Dong case is larger than the others, which is the reflection of Dong catchment's arid climate. The percolation in Jinhua case is larger than the percolation in Xiang case, because the slope in Jinhua catchment is larger." (Section 4.2, line 416 to 418)

Technical corrections:

1. L26 and elsewhere: "studies" instead of "literatures".

*Response:*

Thanks for the reviewer's kind and careful correction. The corrections have been accomplished in the revised manuscript.

2. L26: "indices" instead of 'indexes".

*Response:*

Thanks for the reviewer's kind and careful correction. The corrections have been accomplished in our revised manuscript.

3. L38: ": : : the hydrological model shall be able to"; what is meant by this sentence? How does it relate to the previous sentence?

*Response:*

Thanks for the reviewer's kind correction. The sentence "A good hydrological model shall be able to reproduce the streamflow well in all aspects, which means simulated streamflow series and observed streamflow series have similar Hausdorff dimensions." is now changed to "That means, to reproduce all characteristics of observed streamflow, simulated streamflow and observed streamflow should have similar Hausdorff dimensions, as well as other traditional metrics." (line 101 to 103)

4. L41: "hydrological" instead of "hydrology".

*Response:*

Thanks for the reviewer's kind and careful correction. The corrections have been accomplished in the revised manuscript.

5. L50: "interest" instead of "interests".

*Response:*

Thanks for the reviewer's kind and careful correction. The corrections have been accomplished in our revised manuscript.

6. L57: What kind of loss function is referred to here?

*Response:*

The loss functions referred are listed following this sentence. "For example, Hao et al. (2013) proposed a method based on entropy theory for constructing the bivariate distribution of drought duration and severity with different marginal distribution forms. Pechlivanidis et al. (2014) combined conditioned entropy difference metric and Kling-Gupta efficiency for multi-objective calibration of hydrologic models."

To avoid misleading, this sentence "Studies also used methods of loss function in model calibration." is removed.

7. L90-92: What is meant with this sentence? Do you have an example of a situation where the individual data points are well simulated but physical behaviour of the model (for a particular catchment) is not realistic?

*Response:*

Thanks for the reviewer's kind correction. For now, we don't have proper example yet. Even if we have good examples, it's not appropriate to take too much space in this manuscript. Therefore, we removed this sentence.

8. L110-111: What do the authors mean with 'dependent significances theoretically'?

*Response:*

Thanks for the reviewer's kind correction. We changed the sentence as follows:

"The difference of these dimensions is the calculation scheme of fractal dimensions, and they are numerically related and theoretically dependent."

9. L120: What is meant with 'classical criterion controlling water budget'? A calibration criterion which compares observed and simulated water balances?

*Response:*

Thanks for the reviewer's kind correction. We changed this sentence as follows:

"Therefore, on the basis of classical criteria who compare observed and simulated water balances, the Hausdorff dimension can offer useful insight into mechanisms controlling the extreme hydrological events (including floods, droughts and low flows)." (line 95 to 98)

10. L130: What is the meaning of the 'delta' symbol?

*Response:*

Thanks for the reviewer's kind correction. $\delta$ is the resolution used to calculate Hausdorff dimension in our manuscript. We notice that it's where the first time symbol $\delta$ appears. So we changed this sentence to:

"The value of Hausdorff dimension of the same time series may be different for different resolutions."

11. L132: This part of the sentence is not clear and is an example which needs reformulation.

*Response:*

Thanks for the reviewer's kind correction. We rewrote the sentence as:

"The Hausdorff dimension of joint data series (also called as joint multifractal spectrum) verifies the freezing-thawing process of soil moisture in a quantitative and solid way which unfolds the complex nonlinear relationship among three hydrological variables (Bai et al., 2019)."

12. L209: It would be better to use the same colours for the three DEMs.

*Response:*

Thanks for the reviewer's kind correction. We have changed the DEMs using the same colors.

[Figure]

**Figure 3: DEM of study areas.**

13. **L265:** What is the E-RD calibration strategy? Probably here the explanation from sub-section 2.4 can be used.

*Response:*

Thanks for the reviewer's kind correction. Now we put the description of calibration strategy here (section 2.1 in our revised manuscript).

14. **L266:** "corresponding" instead of "correspondent".

*Response:*

Thanks for the reviewer's kind correction. The corrections have been accomplished in the revised manuscript.

15. **L276:** Do you have a reference for 'distance correlation'?

*Response:*

Thanks for the reviewer's kind correction. We have added the reference for "distance

correlation":

"The distance correlation, as a multivariate measure of dependence, calculates the correlation of distances between points to means. The distance correlation is believed to have better performance when solving problems with non-linear data or extreme values (Székely et al., 2007)." (line 253 to 255)

16. L295: "1.0" instead of "$1+2.8 \times 10-12$".

*Response:*

Thanks for the reviewer's kind correction. The corrections have been accomplished in the revised manuscript.

17. L298: What is the meaning of 'significant' here?

*Response:*

Thanks for the reviewer's kind correction. We changed this sentence to:

"According to relevant studies, the biggest difference of Hausdorff dimension of data of the same type is smaller than 0.25 (Hurst, 1951; Rubalcaba, 1997; Meseguer-Ruiz et al., 2019), which indicates the ranges of $RD$ aforementioned present the significant difference of simulated streamflow from the aspect of fractal." (line 274 to 277)

18. L298-299: And when RD is smaller than 1? In this study RD often is lower than 1 and sometime only slightly higher than 1. Why is this the case?

*Response:*

Thanks for the reviewer's kind correction. We made more explanation as:

"In this study, $RD$ is often lower than 1 and sometime only slightly higher than 1, which agrees with the smooth hydrograph and simple structure of HBV model." (line 277 to 279)

19. L314: Xiang catchment has a different (wrong?) x-axis.

*Response:*

Thanks for the reviewer's kind correction. The x-axis of Xiang catchment in Fig. 4 is

correct.

20. L318: Are the best RD, best E and largest RD selected from the Pareto front?

*Response:*

Thanks for the reviewer's kind correction. Yes, the best RD, best E and largest RD are selected from the Pareto front.

21. L329-330: This sentence is not clear.

*Response:*

Thanks for the reviewer's kind correction. We rewrote this sentence as:

"In this study, the calculations of two metrics ($RD$ and $E$) are totally different, and the results of multi-objective calibration also show that the significant change of $RD$ only leads to minor difference of $E$ (see Fig. 4)." (line 313 to 315)

22. L359: "front" instead of "frontier".

*Response:*

Thanks for the reviewer's kind correction. The corrections have been accomplished in the revised manuscript.

23. L377-378: The ranges of the parameters in Table 3 need units. In addition, the ranges for parameter BETA are very small and all below 1. How do these ranges compare with recommended ranges from the literature?

*Response:*

Thanks for the reviewer's kind correction. In the manual of HBV, the ranges of BETA are not determined (HBV light version 2, user's manual). In some studies, the range of BETA also included the 0 to 1 part (Dakhlaoui et al., 2012). And the calibrated values of BETA in our study are related with good performance of hydrological model. Thus, we believe the values are reasonable.

Dakhlaoui, H., Bargaoui, Z., & Bárdossy, A. (2012). Toward a more efficient calibration schema for HBV rainfall–runoff model. Journal of Hydrology, 444, 161-179.

24. L386: The first part of this line is not correct; the correlation for the Dong catchment is smaller than 0.74.

*Response:*

Thanks for the reviewer's kind correction. We now removed this sentence.

25. L391-392: Please include a unit for KF in Fig. 8 and switch the axes; i.e. RD is the dependent variable and the parameters are the independent variables. This also applies to other figures with RD as a function of parameter values.

*Response:*

Thanks for the reviewer's kind correction. The corrections have been accomplished in the revised manuscript.

26. L409-410: What is the meaning of the letters A, B, D, E, G and H?

*Response:*

Thanks for the reviewer's kind correction. A, B, D, E, G and H are used to be the marks of selected models. We removed all marks of selected models and used "best RD", "best E" and "largest RD" to present selected models.